# Posture-Informed Muscular Force Learning for Robust Hand Pressure Estimation

**Kyungjin Seo**[1*]**, Junghoon Seo**[1*]**, Hanseok Jeong**[2]**, Sangpil Kim**[3]**, Sang Ho Yoon**[1,2]

[1] Graduate School of Culture Technology, KAIST, South Korea
[2] Graduate School of Metaverse, KAIST, South Korea
[3] Department of Artificial Intelligence, Korea University, South Korea

## Abstract

We present PiMForce, a novel framework that enhances hand pressure estimation by leveraging 3D hand posture information to augment forearm surface electromyography (sEMG) signals. Our approach utilizes detailed spatial information from 3D hand poses in conjunction with dynamic muscle activity from sEMG to enable accurate and robust whole-hand pressure measurements under diverse hand-object interactions. We also developed a multimodal data collection system that combines a pressure glove, an sEMG armband, and a markerless finger-tracking module. We created a comprehensive dataset from 21 participants, capturing synchronized data of hand posture, sEMG signals, and exerted hand pressure across various hand postures and hand-object interaction scenarios using our collection system. Our framework enables precise hand pressure estimation in complex and natural interaction scenarios. Our approach substantially mitigates the limitations of traditional sEMG-based or vision-based methods by integrating 3D hand posture information with sEMG signals. Video demos, data, and code are available online.[1]

## 1 Introduction

Hands are a central tool for humans to interact with the surrounding environment. With the advancement in hand tracking technology, hand inputs, including position, orientation, gesture, and motion, are increasingly used as a primary means of control, especially for emerging interfaces (e.g., augmented/virtual reality and wearables). Using hands as the main interaction medium offers a high level of versatility and flexibility to achieve natural and intuitive interactions.

Recent studies have started to utilize hand pressure information to support hand-based interactions such as touching [1], grasping [2, 3], and pressing [4]. Researchers also utilized hand pressure to provide effective haptic feedback [5] for a more immersive user experience. Furthermore, precise hand pressure measurement becomes essential for real-world applications, including ergonomic evaluation [6], hand rehabilitation [7], and prosthetic hand control [8]. To this end, previous works focus on obtaining real-time and accurate hand pressure information with direct measurement approaches utilizing gloves [9–11] or load cells [12]. However, these approaches require users to be in physical contact by either wearing or holding the device, which hinders natural hand movements or reduces user comfort. Thus, the necessity of direct contact limits the users from performing hand-based interactions in a natural and unrestricted manner.

To this end, non-invasive sensing techniques to estimate exerted hand pressure without embedding sensors on the user's hand have been highlighted. These methods include profiling wrist topography with capacitive sensing [13], multiple pressure sensing from the wrist [14], and electromyography

---

[*]Both authors contributed equally to this research.

[1]Project Page: https://hci-tech-lab.github.io/PiMForce/

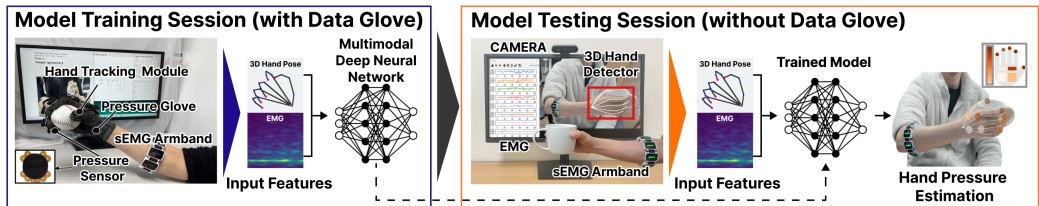

Figure 1: Our sensing framework (PiMForce) leverages 3D hand posture information along with sEMG data to enable a whole-hand pressure estimation during various hand-object interactions. We support real-time pressure estimation on the fingertips and palm regions based on RGB image and sEMG inputs. The intensity of each node's color indicates the pressure level.

from the forearm [2]. Recent works utilized only a single RGB-D [15] or RGB camera [16, 17] with computer vision techniques to estimate pressure exerted by the hand. However, previous methods had limitations, where they could only estimate hand pressure when interacting with plane surfaces or required a line-of-sight view of the hand. With these limitations, it is hard to support natural interaction contexts like working with diverse hand grasps.

In this work, we introduce PiMForce, a novel multimodal sensing framework that enhances real-time hand pressure estimation by leveraging 3D joint information of the hand and forearm sEMG signals, covering the fingertip to the entire palm. As illustrated in Figure 1, our framework addresses previous challenges by integrating detailed spatial information from 3D hand poses with dynamic muscle activity from sEMG measurements. This multimodal sensing integration allows us to estimate subtle and comprehensive hand pressure even under diverse grasps. To validate the proposed model, we built a multimodal hand data collection system and created a dataset from 21 participants. We believe our dataset is the first of its kind containing multimodal sensor signals during hand interactions under various grasps. We demonstrated our work on an off-the-shelf system with a single camera to confirm the accuracy and feasibility of the proposed framework. Our contributions are listed as follows:

- We propose PiMForce, a novel hand pressure estimation framework that enhances sEMG signals by incorporating hand posture information.
- We develop a multimodal hand data collection system with a data collection protocol and create a unique dataset containing simultaneous hand pressure, hand posture, and surface electromyography signals.
- Evaluation and analysis of experiments demonstrate the improved performance of our approach, showing its consistent superiority over existing sEMG-based and vision-based methods.

## 2 Related Works

### 2.1 Vision-based Hand Pressure Estimation

Previous works explored the interaction between the hand and objects by observing the movement and rotation of the object over time to estimate the pressure exerted by hands. By determining the pressure required to produce these observed changes, the model estimated the aggregate pressure applied by the hand [18–20]. These approaches enabled the pressure estimation to act upon concealed or non-visible areas where direct visibility of the hand in contact with an object is absent. Still, previous approaches had limitations where interacting with immovable objects would not work due to the absence of dynamic interaction indicators.

Researchers also looked into different visual indicators like color changes in fingertips, which represent fluctuation of blood circulation within the fingertips [21, 22] or compression of skin tissues [23, 24]. These physiological behaviors served as indicators of the exerted hand pressure. Moreover, examining shadows cast during hand-object interaction provided further insight into the spatial relationship and dynamics of force between them [25–27].

Recent works further advanced the existing visual indicator approach where they use a hand image captured by a single camera at a distance to estimate the hand pressure [16, 28, 17, 29]. They employed a deep learning model that facilitates the understanding of visual cues to estimate accurate

Table 1: Comparison with the previous datasets for hand contact and pressure estimation. We modified and updated the table from [17]. *NA* refers to 'not available'.

| Dataset | Input Modality | Frames | Participants | Contact / Pressure Source | Pressure | Pose | Whole Hand | Natural Objects |
|---|---|---|---|---|---|---|---|---|
| OakInk [47] | RGBD | 230k | 12 | Inferred from pose | × | ✓ | ✓ | ✓ |
| DexYCB [44] | RGBD | 582k | 10 | Inferred from pose | × | ✓ | ✓ | ✓ |
| HO-3D [45] | RGBD | 78k | 10 | Inferred from pose | × | ✓ | ✓ | ✓ |
| GRAB [46] | Pose | 1.6M | 10 | Inferred from pose | × | ✓ | ✓ | × |
| ContactPose [50] | RGBD | 3.0M | 50 | Thermal imprint | × | ✓ | ✓ | × |
| PressureVisionDB [28] | RGB | 3.0M | 36 | Pressure Pad | ✓ | × | ✓ | × |
| ContactLabelDB [17] | RGB | 2.9M | 51 | Pressure Pad | ✓ | × | × | ✓ |
| Force-Aware Interface [4] | sEMG | 17.8M | 9 | Pressure Pad | ✓ | × | × | × |
| HDsEMG [48] | sEMG | 67.6M | 20 | Custom-made Device | ✓ | × | × | × |
| ActionSense [49] | *NA* | 9.42M | 10 | Pressure Glove | ✓ | ✓ | △ | ✓ |
| **Ours** | Pose+sEMG | 83.2M | 21 | Pressure Glove | ✓ | ✓ | ✓ | ✓ |

hand pressure in an end-to-end fashion. However, previous works require a high-quality whole-hand image without occlusion since they rely on visual indicators for the estimation. In our work, we utilize multimodal inputs, including vision-driven 3D hand posture information and wearable-based muscle activation signals, to enhance the estimation of robust hand pressure under various hand-object interaction contexts.

## 2.2 Wearable-based Hand Pressure Estimation

Researchers have used forearm/wrist surface electromyography (sEMG) sensors to acquire finger muscle activation information. Here, the sensor captured a train of neuron impulses propagated through the arms from the forearm or wrist [30, 31]. Previously, researchers used sEMG sensors to estimate various types of hand-related force/pressure, including force/pressure from gripping [32–36] and fingertip [4, 37–41]. Recent works also enabled the estimation of hand pressure along with hand gesture recognition using sEMG signals [8, 42]. However, previous works only dealt with a limited set of discrete hand poses [43]. Moreover, an issue existed with using sEMG signals for complex hand interactions where similar muscle activation signal behaviors were observed across different hand poses. This could easily confuse the model and lead to false behavior. In this work, we train the model with 3D hand posture to encode distinctive hand pose information alongside sEMG signals. This integration forms a robust and accurate hand pressure estimation framework for similar muscle activation behaviors but different hand poses. It is worth emphasizing that this study is the first to incorporate hand posture information for hand pressure estimation using forearm-worn sEMG. Previous studies primarily focused on estimating pressure at the fingertip or on a single gripping force, but our approach expands this to encompass the whole hand.

## 2.3 Datasets for Hand Pressure Estimation

In the computer vision and machine learning community, researchers have formed various types of hand-object interaction datasets for hand pressure estimation. These vision-based datasets collected rich visual and pose information for hand-object interactions, capturing everything from object affordances to whole-body grasps [44, 28, 17, 45–47]. On the other hand, sEMG-based datasets have also been proposed and used to estimate hand pressure for AR/VR or prosthetic robotic arm control applications [48, 4]. A highly relevant work is the ActionSense dataset [49], which focuses on capturing multimodal data of human activities in a kitchen environment using wearable sensors. However, while ActionSense provides a valuable resource for understanding general kitchen activities, our work focuses on the utilization of 3D hand posture in muscular force learning for understanding hand pressure estimation. Furthermore, the temporal resolution of EMG data and the spatial resolution of hand pressure in ActionSense are substantially lower compared to ours, making it challenging to utilize rich sensor input and output effectively. Still, the dataset containing both rich visual and physiological information is missing.

In this work, we attempt to set up a new multimodal dataset that contains 3D hand pose information, sEMG signals, and ground truth measurement of hand pressure as shown in Table 1. Our work provides a holistic view of whole-hand dynamics during various hand-object interactions. This integration enables continuous and comprehensive pressure estimation across the whole palm, addressing the limitations of previous datasets that either infer pressure from visual cues or measure it in isolation.

Our dataset not only captures the subtle interplay between visual and tactile information, but also increases the potential candidates for input features for accurate and robust hand pressure estimation.

# 3 Building Multimodal Dataset: Posture, Electromyography, and Pressure

To capture multimodal data with various hand-object interactions, we integrated and customized existing hardware, including a pressure glove, an armband with 8-channel sEMG sensors, and a markerless finger tracking module. Figure 7 in supplementary material showcases our data collection setup to capture real-time and synchronous multimodal data, including 3D hand posture, sEMG signals, and exerted hand pressure. More detailed information about the hardware, defined hand postures, data collection protocol, and data processing can be found in Section B.

## 3.1 Data Collection Setup

**Pressure Glove.** To capture pressure exerted from the hand, we developed a customized pressure glove using a single 65-node pressure sensing glove (TactileGlove, Pressure Profile Systems) attached with a pressure sensor (RA18DIY, Marveldex) at each fingertip. We added flexible sensors to the fingertips to address missed readings when the pressures were exerted on the edge of the fingertip. Our pressure glove supports pressure readings up to 55 N/cm$^2$ with a sampling rate of 40 Hz.

**3D Hand Pose.** Recent common approaches to obtaining ground truth 3D hand pose [51] involve using multiple RGB cameras to derive 2D hand poses from each camera, followed by triangulation [52, 53] or hand template fitting [44, 45, 54]. However, these methods are infeasible when a pressure data glove is worn, as the glove obscures the hand, hindering accurate hand pose estimation from RGB images. To acquire accurate 3D hand pose information under this constraint, we employed a magnetic sensing-based markerless finger tracking module (Quantum Mocap Metaglove, Manus). The module provides each finger's 3D position and 3-axis joint angles with a sample rate of 120 Hz and less than 5-millisecond latency. We attached the finger-tracking module to the pressure glove to capture exerted hand pressure and 3D hand pose data simultaneously.

**8-Channel sEMG Armband.** We used 8 sEMG sensors (Trigno Avanti, Delsys) and installed sensors into a customized armband made with semi-flexible material (TPU 95A) to ensure electrode contact for various sizes of forearms. Our system captures muscle action potentials with a sampling rate of 2,000 Hz.

**Multimodal Data Synchronization.** Before training the multimodal dataset, we employed a linear interpolation approach to synchronize high frame rate readings (sEMG) with low frame rate data (3D hand pose). We first joined the data from matched time points based on the collection time of the sEMG data and interpolated missing values of the hand pose data linearly. We applied the same approach to the pressure glove, where we synchronized high frame rate sensors (pressure sensors) with the low frame rate sensing glove (TactileGlove). Then, we adopted a nearest-neighbor-based interpolation to synchronize 3D hand pose and sEMG data with hand pressure [4].

## 3.2 Data Collection Procedure

With IRB approval, a total of 21 right-handed participants took part in this study. Of these, 17 were male (81%) and 4 were female (19%). The participants' ages ranged from 20 to 32 years, with a mean age of 24.3 years (SD = 3.9). To ensure good quality hand pressure data using our glove, we chose participants with hand sizes greater than 180 mm. Prior to participation, all participants were provided with a detailed information sheet outlining the purpose, procedures, and potential risks of the study. We obtained written informed consent from each participant, ensuring they understood the nature of the data being collected, their right to withdraw at any time, and the measures taken to ensure data privacy. We equipped participants with our multimodal glove and an 8-channel sEMG armband. Following initial calibration to compensate for each user's hand size, participants performed 22 distinctive hand-object interactions for the data collection task (Figure 8 in supplementary material).

Our hand-object interactions consist of 7 hand-plane interactions, 5 pinch interactions, and 10 distinctive hand grasps. We included the same hand-plane and pinch interactions from recent hand pressure estimation work [4] while adding palm-pressing motion. In terms of hand grasps selection,

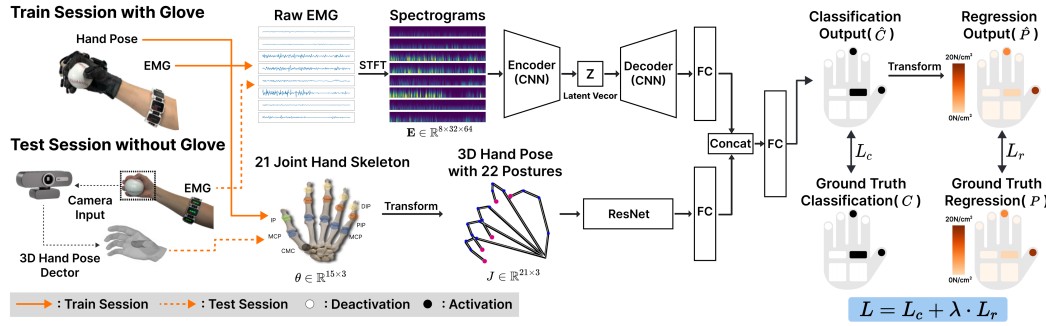

Figure 2: Our multimodal hand pressure estimation architecture enhances sEMG data by embedding 3D hand pose information. We train the model using a classification-regression joint loss to improve hand pressure estimation.

we selected 10 grasps from representative 33 grasp types [55] based on the clustering of palm pressure distribution similarity among grasp taxonomies [56]. We collected 1,980 seconds of synchronized multimodal data per participant (30 seconds $\times$ 22 hand-object interactions $\times$ 3 sessions). Here, we used data from 2 sessions for training while the remaining session was reserved for evaluation. We provided sufficient rest periods between each trial to prevent muscle fatigue, ensuring the quality of collected data. All collected data was anonymized, complying with relevant data privacy regulations.

# 4 Method

## 4.1 Overview

The rationale behind using sEMG to estimate the pressure exerted by the hand lies in the direct relationship between muscle electrical activity and the pressure generated during hand interactions [57–59]. When muscles contract for movement, they generate bioelectric signals that can be captured by EMG sensors [60]. This indicates that sEMG signals have the potential to estimate the pressure exerted by the hand. The ability to decode sEMG signals from forearm muscles generated by finger-level movements will set a robust foundation for understanding complex hand interactions. However, the pressure exerted by the hand cannot be solely represented with muscle activation information. The main reason is that the distribution of hand pressure varies according to different hand postures. For example, similar sEMG patterns may be generated by different hand pressures depending on the related hand postures or grasps [43, 61]. This highlights the importance of considering hand posture information along with sEMG signals to estimate the exerted hand pressure precisely. Section C.1 in supplementary material addresses the specific empirical observation for this motivation.

To address these issues, we enhance sEMG signals by leveraging 3D hand posture information. By integrating inputs from forearm-worn sEMG sensors with 3D hand pose information derived from an RGB image, we observe improvements in the accuracy of hand pressure estimation. Our multimodal approach (Figure 2) leverages the strengths of both hand posture and muscle activations, offering a comprehensive understanding of hand dynamics for whole-hand pressure estimation. Refer to Section C for more detailed information about the model architecture, training, and inference.

## 4.2 3D Hand Pose and sEMG Feature Extractions

To verify the validity of our framework, we devise a deep neural network model to effectively utilize the obtained multi-modalities. We represent the model as $f$, where the 3D hand pose is denoted by $H$ and the sEMG signal by $E$. The classification and regression targets for pressure are represented as $C$ and $P$, respectively. If the model outputs for pressure classification and regression are indicated, they are denoted by $\hat{C}$ and $\hat{P}$. The feature extractor for sEMG data is $f_{\text{EMG}}$, and for hand pose, it is $f_{\text{hand}}$. The overall model $f$ comprises $f_{\text{EMG}}$, $f_{\text{hand}}$, and a pressure predictor $f_{\text{pred}}$ that takes the features extracted from hand pose and sEMG to perform pressure classification and regression.

**Feature Extraction from sEMG signals.** The sEMG data generally encounters measurement noises, including powerline noise and electromagnetic artifacts. To mitigate these issues, we utilize the short-time Fourier transform (STFT) to convert sEMG time-domain signals into spectrograms, represented

as $\mathbf{E} \in \mathbb{R}^{8 \times 32 \times 64}$, isolating high-frequency noise and facilitating the application of convolutional neural network (CNN) models for feature extraction. We employed a 2D encoder-decoder model to extract features from the 2D sEMG signals. The encoder-decoder model processes the data and then flattens the output, which is subsequently transformed through a fully connected (FC) layer into a 512-dimensional feature vector.

**Feature Extraction from 3D Hand Pose.** The hand model in our study is represented as a kinematic tree with 15 joint angles $\theta \in \mathbb{R}^{15 \times 3}$, similar to the pose parameters in MANO and its variants [62–65]. To handle this skeleton-based representation simply like PoseConv3D [66], we adopt a 3D ResNet [67] to process 3D heatmap volumes of hand joints, transforming $\theta$ into 21 3D hand joints $J \in \mathbb{R}^{21 \times 3}$ through the hand skeleton model's forward kinematics. These 3D hand joints are then converted into 3D heatmap volumes $H \in [0, 1]^{21 \times H \times W \times D}$, with $H$, $W$, and $D$ all set to 48, for processing by the 3D ResNet. Unlike the sequential representation of 2D heatmaps in PoseConv3D, our approach uses a single timestep 3D joint representation. After processing through the 3D ResNet, the hand pose feature is flattened and transformed into a 512-dimensional vector using an FC layer, with the 3D ResNet34 model being the model of choice for this operation.

**Feature Fusion and Estimation.** The extracted sEMG feature $f_{\text{EMG}}(E)$ and hand pose feature $f_{\text{hand}}(H)$ are concatenated to form a 1024-dimensional joint feature vector. This vector is then passed through two FC layers, mapping to 256-dimensional features, followed by a 1-D batch normalization and ReLU non-linearity. Finally, the last FC layer with Sigmoid activation function maps this to $I$-dimensional output, producing $\hat{C} = f(E, H) = f_{\text{pred}}(f_{\text{EMG}}(E), f_{\text{hand}}(H)) \in [0, 1]$. The predicted pressure $\hat{P}$ is defined as $\hat{P} = 2P_{\text{max}} \cdot \text{ReLU}(\hat{C} - 0.5) \in [0, P_{\text{max}}]$, thus completing the feature fusion and pressure prediction.

## 4.3   Joint Training of Classification and Regression

The objective function comprises two key components: a classification loss $L_c$ and a regression loss $L_r$. The classification loss is designed to accurately identify whether any pressure is exerted by a particular region of the hand (i.e., fingertips or palm areas shown in Figure 5 of Section B.1.1), using a cross-entropy loss to distinguish between pressure and no-pressure instances for each hand region $i$:

$$L_c = \frac{1}{I} \sum_{i=1}^{I} C_i \cdot \log \hat{C}_i + (1 - C_i) \cdot \log(1 - \hat{C}_i), \tag{1}$$

where $C_i$ is the ground-truth label for region $i$, and $\hat{C}_i$ is the predicted probability of pressure application. Here, $C_i = 1$ indicates the presence of pressure in the $i$th region, while $C_i = 0$ indicates its absence. In contrast, the regression loss, targets the accurate quantification of pressure levels using an $L_2$ loss to minimize the difference between the predicted and actual pressure values:

$$L_r = \frac{1}{I} \sum_{i=1}^{I} \|\hat{P}_i - P_i\|^2, \tag{2}$$

where $\hat{P}_i$ represents the model's predicted pressure for region $i$ and $P_i$ is the corresponding actual pressure. Our dataset contains pressures from 0$\sim$20 N. Therefore, our model predicts pressure values in [0, 20], organized by $\hat{P}_{max}$, the maximum of the pressure. To integrate these two aspects into a unified training objective, we introduce a balancing hyper-parameter $\lambda$, resulting in a combined loss function: $L = L_c + \lambda \cdot L_r$. This composite loss enables our model to not only discern the presence of pressure but also quantify its magnitude accurately.

## 4.4   Estimation without the Glove

For the training phase, we employed data acquired from our data collection system to ensure the accurate capture of exerted hand pressure and 3D hand pose data. However, during the inference phase, our framework exploits off-the-shelf hand pose detectors [68–70], which extract 3D hand pose from RGB or RGB+D inputs. These detectors can been chosen for their high accuracy and robustness in various conditions, ensuring reliable performance during inference. Thus, users can interact with external objects using their bare hands, without the need for additional hand-worn equipment. This approach ensures our model's practical applicability in real-world scenarios, prioritizing user convenience and natural interaction. By leveraging readily available technology, we make it easier

Table 2: Performance among comparative models on evaluation metrics.

| Method | $R^2$ | NRMSE | Accuracy |
|---|---|---|---|
| sEMG Only [4] | $83.49 \pm 16.40\%$ | $8.07 \pm 2.62\%$ | $77.83 \pm 11.56\%$ |
| 3D Hand Posture Only | $66.32 \pm 37.01\%$ | $11.57 \pm 3.95$ | $70.08 \pm 13.09\%$ |
| sEMG + Hand Angles | $84.22 \pm 17.11\%$ | $7.89 \pm 2.61\%$ | $78.22 \pm 10.57\%$ |
| PiMForce (Ours) | $\mathbf{88.86} \pm 11.92\%$ | $\mathbf{6.65} \pm 2.11\%$ | $\mathbf{83.17} \pm 9.38\%$ |

for users to adopt our system in everyday applications. Refer to Section B.4 for details on how we canonicalized 3D hand pose information extracted from RGB images for our model input.

## 5 Experiments

In Sections 5.2.1 and 5.2.2, where ground truth hand pressure is necessary, data was collected while participants wore the pressure glove, and hand postures were obtained from the data glove. We also conducted qualitative evaluations (Section 5.2.3 and the demo video) without ground truth, where data was collected without any gloves, and hand postures were inferred solely from RGB images using an off-the-shelf hand pose detector. For this purpose, we employed the pre-trained Attention Collaboration-based Regressor [69], which has demonstrated superior performance with a mean per joint position error (MPJPE) of approximately 8mm for reconstructing hand poses from a single RGB camera. The high accuracy ensures the reliability of our hand posture inferences in qualitative assessments. To assess our model's performance, we utilize three metrics: Coefficient of Determination ($R^2$), Normalized Root Mean Squared Error (NRMSE), and classification accuracy. The exact definitions and explanations of evaluation metrics can be found in Section D.2. Refer to Section D.3 and D.4 for additional quantitative and qualitative restuls, respectively.

### 5.1 Comparative Methods

This study compares the proposed model against several baseline and state-of-the-art methods to validate its effectiveness in whole-hand pressure estimation. To ensure a fair comparison, we selected methods that quantitatively measure the pressure applied by the hand, rather than solely identifying hand contact. Detailed information about the implementation of comparative methods can be found in Section D.1. The methods included in the comparison are:

**sEMG Only Model [4].** An sEMG-based approach decodes finger-wise forces in real-time, demonstrating the potential of muscle activation patterns in informing hand activities. This method emphasizes using electromyography sensors for understanding complex hand dynamics but does not incorporate hand posture information.

**3D Hand Posture Only Model.** A variation of our proposed framework that solely utilizes 3D hand posture for pressure estimation, omitting the sEMG signal input. This model tests the efficacy of hand posture information in isolation.

**sEMG + Hand Angles Model.** This model represents a variation of our proposed framework, where instead of utilizing the 3D representation $H$ for hand pose, it employs the angular representation $\theta$ of hand joints as the input to the hand pose feature extractor $f_{\text{hand}}$. By substituting the 3D hand pose with direct angle measurements of hand joints, this baseline aims to highlight the benefits of using a 3D representation for hand pose in multimodal sensing.

**PressureVision++ [17].** This vision-based deep learning model estimates hand pressure from a single RGB image by identifying visual cues related to hand pressure application, showcasing the use of visual information for pressure estimation without physical FSR sensors.

**PiMForce (Ours).** The comprehensive model enhances sEMG signals by leveraging 3D hand posture information for continuous and detailed pressure estimation across the whole hand. This approach aims to mitigate the limitations of sEMG-based methods by integrating the strengths of both modalities for enhanced pressure prediction accuracy.

Table 3: Cross-user performance on evaluation metrics under whole interaction and posture.

| Method | $R^2$ | NRMSE | MAE | Accuracy |
|---|---|---|---|---|
| **sEMG only [4]** | $47.90 \pm 9.97\%$ | $14.14 \pm 2.41\%$ | $12.24 \pm 2.75\%$ | $57.40 \pm 5.81\%$ |
| **PiMForce (Ours)** | $\mathbf{70.06} \pm 4.02\%$ | $\mathbf{10.70} \pm 1.43\%$ | $\mathbf{8.54} \pm 1.56\%$ | $\mathbf{72.01} \pm 2.86\%$ |

## 5.2 Results

We analyze the performance of our proposed framework in comparison to these methodologies, both quantitatively and qualitatively. Additionally, we investigate the capabilities of our model to accurately estimate hand pressure across a variety of hand postures and parts, providing a thorough assessment of its performance. Our framework enhances sEMG signals by leveraging 3D hand posture information for detailed palm pressure data collection, contrasting with PressureVision++, which relies on visual cues for force estimation. This approach is designed to underscore the distinctive benefits of our multimodal sensing framework in capturing a broad range of hand interactions.

### 5.2.1 Do hand pose and sEMG signals together improve pressure estimation?

Table 2 outlines the performance metrics of various comparative models, including the sEMG Only Model, the 3D Hand Posture Only Model, the sEMG + Hand Angles model, and our model. PiMForce remarkably outperforms the comparative methods, achieving an accuracy of 83.17%, NRMSE of 6.65%, and an $R^2$ value of 88.86%. This demonstrates the comprehensive capability of our model to accurately classify and quantify the pressures exerted by the hand.

The integration of 3D hand posture and sEMG information in our framework shows a clear advantage over approaches relying on a single data modality, as expected. The sEMG Only Model and the 3D Hand Posture Only Model show limited pressure estimation performance when compared to our integrated approach. Interestingly, the improvement in performance with the sEMG + Hand Angles model over the sEMG Only Model is marginal (less than 0.5%p improvements in all metrics). This highlights the importance of incorporating a comprehensive 3D hand posture representation. By embedding comprehensive hand posture knowledge to be used with sEMG data, we develop an effective multimodal approach to capture nuanced variations in hand pressure exerted across different hand regions and postures.

Cross-user performance assesses how well the model performs on data from individuals not included in the training set, which is crucial for real-world applications. As shown in Table 3, our proposed method combining sEMG signals with 3D hand posture data significantly outperforms the sEMG-only baseline across all evaluation metrics in cross-user scenarios. This demonstrates the enhanced generalizability and effectiveness of our approach in estimating hand pressure among different users. To further demonstrate the performance of our model over time, we present Figure 23 in supplementary material, which illustrates the temporal evolution of both ground truth and predicted pressure values for all nine hand regions during consecutive TM-Press and Medium Wrap actions.

### 5.2.2 How does accuracy vary by hand region and posture type?

We delve into the performance of our model across various hand regions and posture types, utilizing data represented in both Table 4 and Figure 3. This analysis highlights the noticeable impact of incorporating 3D hand pose data, particularly noting a greater improvement in hand palm regions (+1.95%p) over fingertips (+1.05%p) compared to the sEMG Only Model. This distinction emphasizes the crucial role of 3D hand pose for accurate pressure estimation in diverse hand postures.

Our findings reveal that the model achieves superior pressure estimation in *Press* and *Pinch* interactions, with classification accuracies surpassing 90% and NRMSE values maintained below 6%. However, it encounters challenges with specific postures such as *Palm-Press*, which, despite a lower classification accuracy of 68.42%, still shows a high regression accuracy of 3.12%. When examining *Grasp* postures, our model shows a slight dip in performance relative to *Press* and *Pinch*, with NRMSE values ranging between 5∼8%. This suggests a moderate pressure estimation capability for these more complex interactions, yet the model consistently maintains a high $R^2$ range of 0.8 to 0.9 across all posture types. This consistent correlation between predicted values and actual pressure measurements highlights the model's ability to maintain high accuracy and reliability across a diverse

Table 4: Performance comparison of hand regions in terms of NRMSE.

| Method | Finger Tip | | | | | | Hand Palm | | | | | Overall Mean |
|---|---|---|---|---|---|---|---|---|---|---|---|---|
| | Thumb | Index | Middle | Ring | Pinky | Mean | Upper Right | Upper Left | Lower Right | Lower Left | Mean | |
| sEMG Only [4] | 9.78 ± 3.45% | 9.26 ± 3.51% | 8.15 ± 3.17% | 6.65 ± 2.51% | 4.69 ± 1.32% | 7.71 ± 2.79% | 8.79 ± 3.88% | 8.43 ± 3.76% | 8.66 ± 3.78% | 7.79 ± 3.98% | 8.42 ± 3.85% | 8.07 ± 3.26% |
| 3D Hand Posture Only | 13.85 ± 5.39% | 13.64 ± 5.39% | 12.18 ± 5.31% | 9.85 ± 4.49% | 6.16 ± 1.73% | 11.13 ± 4.55% | 12.26 ± 6.09% | 10.41 ± 6.30% | 11.47 ± 6.02% | 9.71 ± 6.30% | 10.96 ± 6.18% | 11.05 ± 5.27% |
| sEMG + Hand Angles | 9.54 ± 3.48% | 9.04 ± 3.56% | 7.95 ± 3.21% | 6.48 ± 2.48% | 4.57 ± 1.37% | 7.52 ± 2.82% | 8.56 ± 3.90% | 7.23 ± 4.14% | 7.67 ± 4.22% | 6.67 ± 4.21% | 7.53 ± 4.12% | 7.52 ± 3.40% |
| **PiMForce (Ours)** | **8.04 ±** 2.76% | **7.73 ±** 2.79% | **6.99 ±** 2.61% | **6.01 ±** 2.39% | **4.06 ±** 0.90% | **6.66 ±** 2.29% | **7.23 ±** 3.12% | **6.17 ±** 3.32% | **6.76 ±** 3.13% | **5.73 ±** 3.38% | **6.47 ±** 3.24% | **6.52 ±** 2.71% |

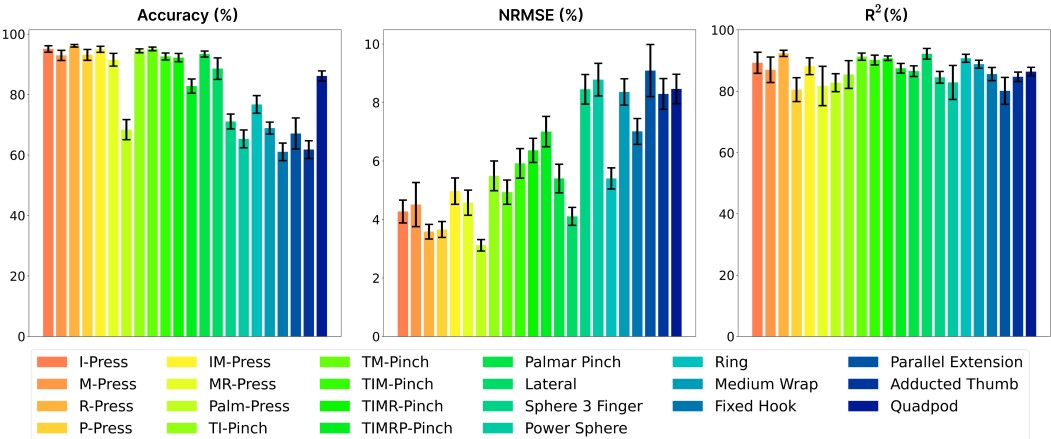

Figure 3: Quantitative evaluation of the user-independent model, showing the posture-wise performance on the estimation of hand pressure. The error bars indicate standard error.

range of hand parts and postures. Specific actions such as *I-Press*, *M-Press*, and *R-Press* exhibit high accuracy and low NRMSE, showing the model's superior performance in simpler press interactions. On the contrary, more complex grasps like *Power Sphere*, *Fixed Hook*, and *Parallel Extension*, showed lower accuracy and higher NRMSE, implying that there remains room for improvement.

### 5.2.3 Can inference succeed with an off-the-shelf hand pose detector?

For practical inference applications without a data glove, solely relying on EMG data and incorporating 3D hand pose information obtained through an off-the-shelf hand pose detector, we demonstrate the adaptability of our model in Figure 4a. This comparison with the vision-based method, PressureVision++, showcases our PiMForce's capability to estimate hand pressures robustly during diverse interactions. During hand-plane interactions, specifically those involving the tip *Press* type posture, both approaches appear to perform well. However, our analysis reveals vulnerabilities in handling more complex *Grasp* and *Pinch* motions when using PressureVision++. Furthermore, PressureVision++ requires complete visibility of all fingers within the camera's view due to the high reliance on visual cues for pressure inference. In contrast, our framework effectively utilizes the estimated hand pose as long as the hand pose information is sufficiently accurate for inference. This capability underscores the practicality of our method, facilitating more natural user interactions with external objects without the constraints of direct visibility or glove use. Figure 4b shows demo video footage illustrating our PiMForce's capability to accurately estimate hand pressure while continuously changing hand posture, pressure levels, and the objects being grasped. This demonstrates the flexibility and reliability of our approach in real-world scenarios.

To further substantiate our model's effectiveness using an off-the-shelf hand pose detector, we conducted a quantitative comparison with PressureVision++, as presented in Table 5. PiMForce demonstrates largely better performance across all fingertips during both plane and pinch interactions, indicating superior performance in estimating hand pressures compared to PressureVision++. This quantitative evaluation confirms that our framework outperforms existing vision-based methods in terms of accuracy and robustness during diverse interactions.

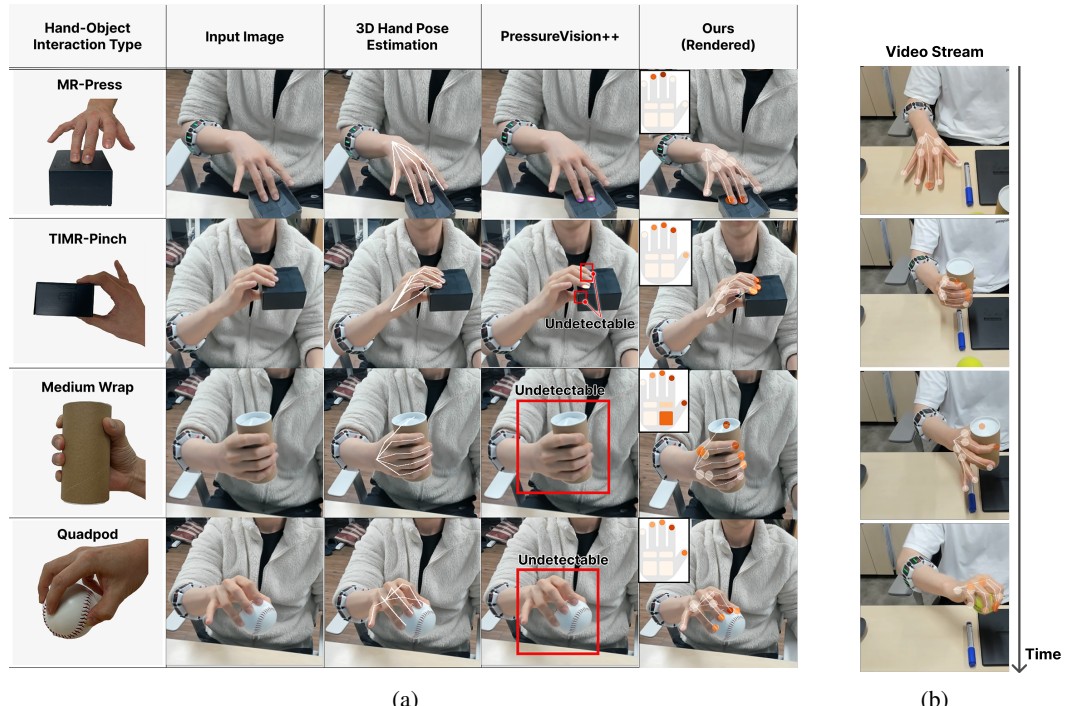

|     |     |
|:---:|:---:|
| (a) | (b) |

Figure 4: **(a)** Qualitative results in the absence of a pressure glove. The 3D Hand Pose Estimation [69] represents 3D hand posture, including hand occlusion, using the 3D hand detector. The Pressure-Vision++ [28] column shows the pressure estimation of fingertips. The red rectangles indicate the instances of pressure estimation failure due to hand occlusion. The proposed multimodal framework shows robust whole-hand pressure estimation for diverse hand-object interactions. **(b)** Illustration of the demo video footage showing robust hand pressure estimation with varying hand postures, pressure levels, and interacting objects.

Table 5: Cross-user performance on evaluation metrics under plane interaction and pinch posture.

| Method | $R^2$ | NRMSE | Accuracy |
|:---:|:---:|:---:|:---:|
| **PressureVision++ [17]** | $40.30 \pm 5.14\%$ | $32.95 \pm 2.02\%$ | $67.90 \pm 3.01\%$ |
| **sEMG only [4]** | $42.13 \pm 6.88\%$ | $12.57 \pm 2.09\%$ | $66.00 \pm 5.84\%$ |
| **PiMForce (Ours)** | $\mathbf{66.71} \pm 4.68\%$ | $\mathbf{9.27} \pm 1.40\%$ | $\mathbf{82.20} \pm 2.42\%$ |

# 6   Conclusion

In this paper, we introduce PiMForce, a pioneering framework for hand pressure estimation by integrating 3D hand posture information with muscle activation signals from forearm-worn sEMG. By embedding 3D hand posture information into a deep neural network, we enable the model to process this data alongside sEMG signals, enhancing its capability to learn complex relationships between muscle activations and hand pressure distributions. This novel approach is the first to combine these modalities, providing a comprehensive analysis of hand dynamics across various interactions. We developed a unique multimodal hand data collection system and protocol, capturing a dataset that includes hand pressure, posture, and electromyography signals. Our method notably improves upon previous techniques, enabling accurate whole-hand pressure estimation through detailed hand posture information. Extensive quantitative and qualitative comparisons demonstrated the consistent superiority of our framework over existing sEMG-based and vision-based methods.

## Acknowledgments and Disclosure of Funding

This work was supported by the National Research Foundation of Korea (NRF) grant funded by the Korea government (MSIT) (No. 2022R1A4A5033689, Contribution Rate: 50% and No. 00210001, Contribution Rate: 25%) and Culture, Sports and Tourism R&D Program through the Korea Creative Content Agency grant funded by the Ministry of Culture, Sports and Tourism in 2023 and 2024 (Project Name: Development of Real-time Virtual Convergence-based Performing Arts Education Platform Technology, Project Number: RS-2023-00219020, Contribution Rate: 15%, Project Name: International Collaborative Research and Global Talent Development for the Development of Copyright Management and Protection Technologies for Generative AI, Project Number: RS-2024-00345025, Contribution Rate: 10%).

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

## A  Overall Structure of Supplementary Material

This supplementary material provides added details to complement the main manuscript. Section B elaborates on the specifics of the multimodal dataset creation. Section C offers an extended description of the methodological framework employed in our study. Section D describes implementation details of comparative methods and showcases additional results to further support the findings reported in the main text. Lastly, we note limitations and future works in Section E.

## B  Detailed Description for the Multimodal Dataset

### B.1  Data Collection Hardware

#### B.1.1  Pressure Sensor Glove Customization

To effectively capture comprehensive data on hand pressure and posture, we modified a tactile glove equipped with 65 pressure sensors by integrating additional fingertip pressure sensors along with a position tracking module, as shown in Figure 5. We customized the design of the glove to focus on the hand pressure measurements across 5 fingertips and 4 palm regions consisting of 16 pressure sensor nodes.To determine the pressure in each region, we use the maximum value among the sensors within that region rather than summing or averaging their readings. This approach accounts for variations in hand size, ensuring that the pressure measurement is not artificially lowered due to inactive sensors that may not be engaged by all participants.

#### B.1.2  Pressure Sensor Characteristics

For fingertip pressure measurements, we utilized a Force-Sensing Resistor (FSR) type pressure sensor (RA18DIY, Marveldex), characterized by a force range of $0 \sim 40$ N/cm$^2$ and a thin profile of 0.7 mm with an 8 mm diameter sensing area. Figure 6 presents the typical response behavior of the FSR, highlighting the force-resistance relationship. In our work, we derived a precise fitting model for the sensor's output through repetitive calibrations using a push-pull gauge. The calibration process involved conducting 30 load-unload cycles (ranging from $0 \sim 30$ N, incremented by 1 N) and meticulously recording the resistance values to ensure accuracy.

### B.2  Defined Hand Postures

For the grasp postures, we carefully selected 10 representative movements from a comprehensive grasp taxonomy presented in recent literature [55]. This taxonomy classifies grasps based on several factors, including opposition type, virtual fingers, grip type, and thumb position, culminating in

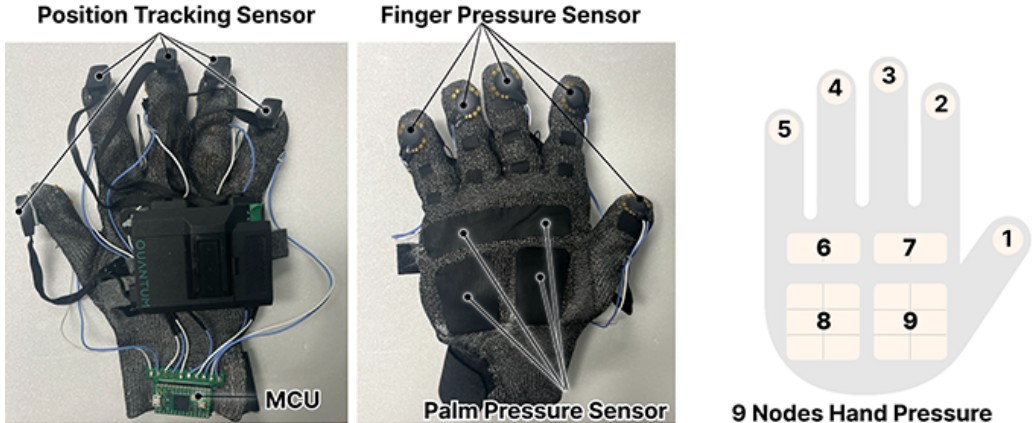

Figure 5: We added position tracking sensors at the knuckles and 5 fingertip pressure sensors. Our glove records occlusion-free hand-tracking data with exerted hand pressures at 9 regions (5 fingertips and 4 palm regions).

Table 6: The detailed data collection protocol for our study, categorizing hand postures into *Plane*, *Pinch*, and *Grasp* postures. It lists each posture's name and abbreviation.

| Type | Posture Name | Abbreviation |
|---|---|---|
| Plane | Index Press | I-Press |
| Plane | Middle Press | M-Press |
| Plane | Ring Press | R-Press |
| Plane | Pinky Press | P-Press |
| Plane | Thumb & Index Press | TI-Press |
| Plane | Index & Middle Press | IM-Press |
| Plane | Palm Press | Palm-Press |
| Pinch | Thumb & Index Pinch | TI-Pinch |
| Pinch | Thumb & Middle Pinch | TM-Pinch |
| Pinch | Thumb & Index & Middle Pinch | TIM-Pinch |
| Pinch | Thumb & Index & Middle & Ring Pinch | TIMR-Pinch |
| Pinch | Thumb & Index & Middle & Ring & Pinky Pinch | TIMRP-Pinch |
| Grasp | Palmar Pinch | — |
| Grasp | Lateral | — |
| Grasp | Sphere 3 Finger | — |
| Grasp | Power Sphere | — |
| Grasp | Ring | — |
| Grasp | Medium Wrap | — |
| Grasp | Fixed Hook | — |
| Grasp | Quadpod | — |
| Grasp | Parallel Extension | — |
| Grasp | Adducted Thumb | — |

33 distinct grasp types. Given the constraints of wearing an EMG armband and the necessity to minimize experimenter fatigue through sufficient rest, it was crucial to narrow down the posture set. Following the clustering of grasp types based on pressure distribution in [56], we opted for ten hand movements that epitomize the diverse range of grasps, as shown in Figure 9. This selection process ensured a manageable yet comprehensive dataset that accurately reflects a wide array of hand-object interactions.

## B.3 Data Collection Protocol

As illustrated in Figure 10, we developed software tools to facilitate user convenience and high-quality data acquisition. These tools provide posture-specific guide images to make it easier for users to follow along, as well as visual feedback that allows monitoring of EMG, pressure, and hand movements in real time. Additionally, they offer information on data collection and rest times for each posture, enabling systematic data gathering with time synchronization.

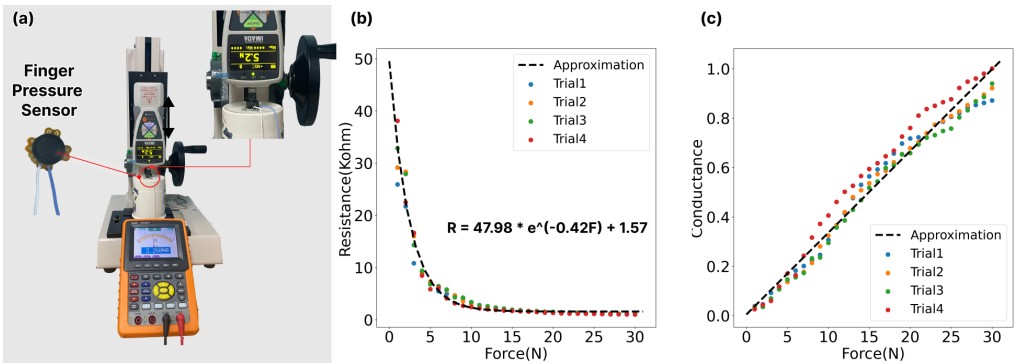

Figure 6: FSR sensor calibration and response curves. **(a)** We calibrated each FSR pressure sensor with the calibration setup using the precise push-pull gauge. **(b) & (c)** The FSR's resistance and conductance characteristics as a function of applied force, respectively, illustrating the sensor's calibration curve derived from multiple trials for data collection.

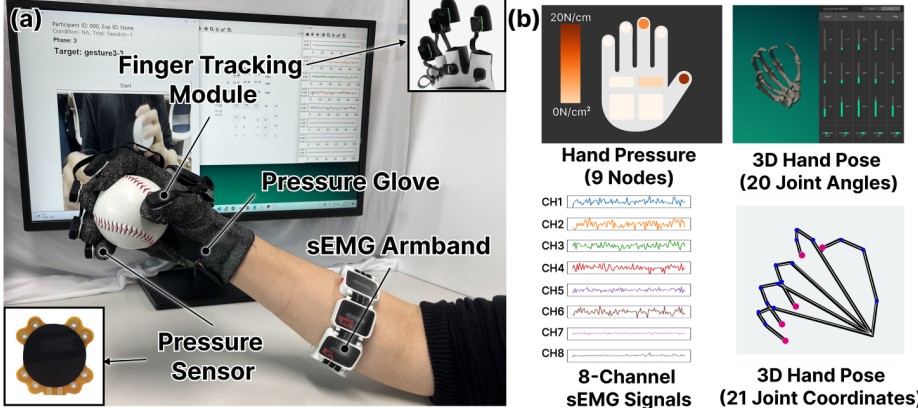

Figure 7: (a) Data collection setup to capture time-synchronized hand pressure, 3D hand pose, and sEMG data. We asked participants to interact with 22 action sets while wearing customized pressure gloves integrated with a finger tracking module and an 8-channel EMG armband. (b) We captured 9-nodes hand pressure values, 8-channel sEMG signals, 20 joint angles, and 21 joint coordinates (computed from joint angles).

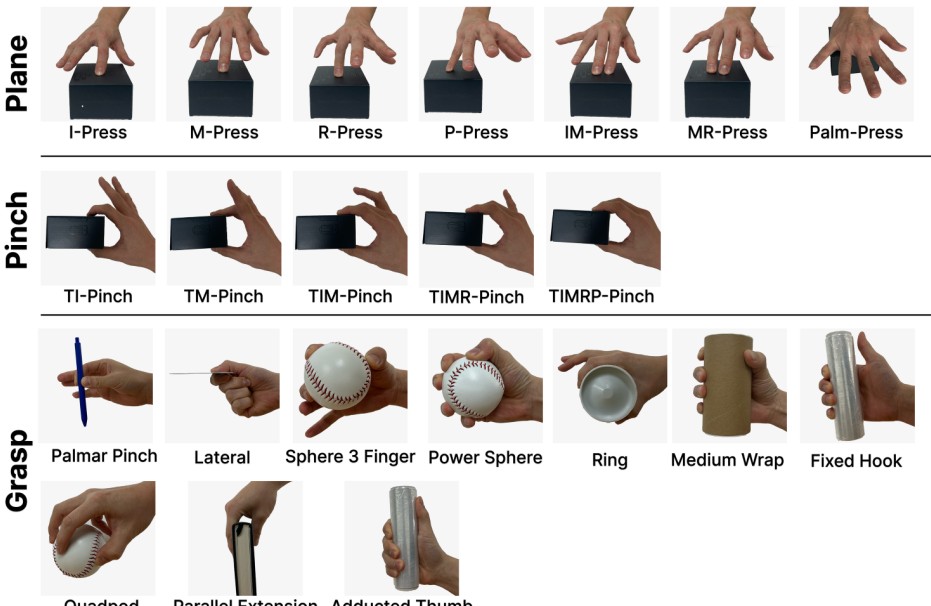

Figure 8: 22 actions executed by the hand while collecting multimodal sensing data. It includes 7 hand-plane interactions, 5 pinch actions, and 10 hand-object interactions selected from hand grasp taxonomy. Capital letters before the hyphen, namely T, I, M, R, and P, stand for thumb, index finger, middle finger, ring finger, and pinky finger, respectively.

Using these data collection tools, we gathered data from each participant for the 22 postures listed in Table 6, repeating each posture a total of three times. Each posture was collected for a duration of 30 seconds, with participants instructed to repeat the given posture 12 to 15 times. For *Plane*-type postures, participants were instructed to apply and then release force against a flat surface. For *Pinch* and *Grasp* postures, they were directed to fix an object with their left hand and then apply and release force with their right hand. To prevent the accumulation of fatigue in the muscles due to EMG, we incorporated a 10-second rest period after each posture, and after completing all 22 postures, participants were given a 10-minute break to ensure adequate rest.

Muscle fatigue, characterized by a decline in a muscle's ability to generate force, occurs when a muscle is repeatedly contracted or held in a sustained contraction over an extended period. As muscles fatigue, their electrical activity changes, manifesting as increased signal amplitude (root mean square) and shifted frequency content [71]. These alterations can negatively impact the quality of sEMG data

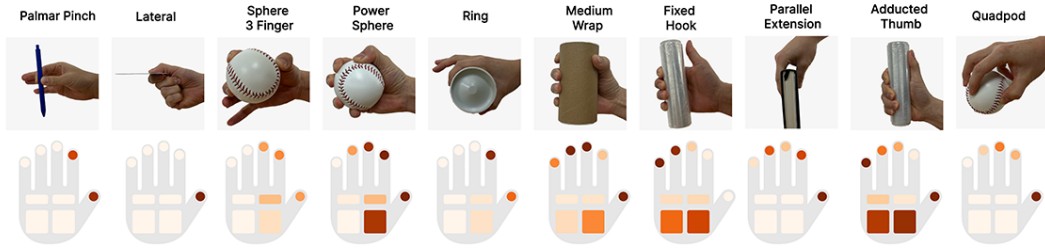

## Grasp Actions

| | Power | | | | | | Intermediate | | | Precision | | | | |
|---|---|---|---|---|---|---|---|---|---|---|---|---|---|---|
| | **Palm** | | **Pad** | | | | **Side** | | | **Pad** | | | | **Side** |
| | 3-5 | 2-5 | 2 | 2-3 | 2-4 | 2-5 | 2 | 3 | 3-4 | 2 | 2-3 | 2-4 | 2-5 | 3 |
| **Thumb Abducted** | | Large Diameter, Power Disk, Small Diameter, Power Sphere, Medium Wrap | Ring | Sphere 3 Finger | Prismatic 4 Finger, Sphere 4 Finger | Distal Type | Adduction Grip | | Tripod Variation | Palmar Pinch, Tip Pinch, Inferior Pincer | Prismatic 2 Finger, Tripod | Prismatic 3 Finger, Quadpod | Prismatic 4 Finger, Precision Disk, Precision Sphere | Writing Tripod |
| **Thumb Adducted** | Index Finger Extension | Adducted Thumb, Palmar, Light Tool, Fixed Hook | | | | | Lateral, Stick, Ventral | Lateral Tripod | | | | | Parallel Extension | |

Figure 9: Representative grasp postures selected for data collection. This illustrates the ten grasp postures chosen for the study, reflecting a wide range of hand interactions. The postures are categorized by opposition type, virtual finger usage, grip type, and thumb position. The color-coded diagrams above the images indicate the pressure points for each grasp, corresponding to the regions of the hand engaged during the posture.

used for hand pressure estimation, making it more challenging for the model to accurately distinguish between different hand postures and pressure levels. Fatigue leads to less consistent muscle activation patterns, complicating the model's ability to learn reliable relationships between sEMG signals and exerted pressure [43]. While muscle fatigue is an important consideration in EMG-based control, addressing it falls outside the scope of this study. To minimize its effects on our results, we ensured sufficient rest periods between data collection trials during both training and testing phases.

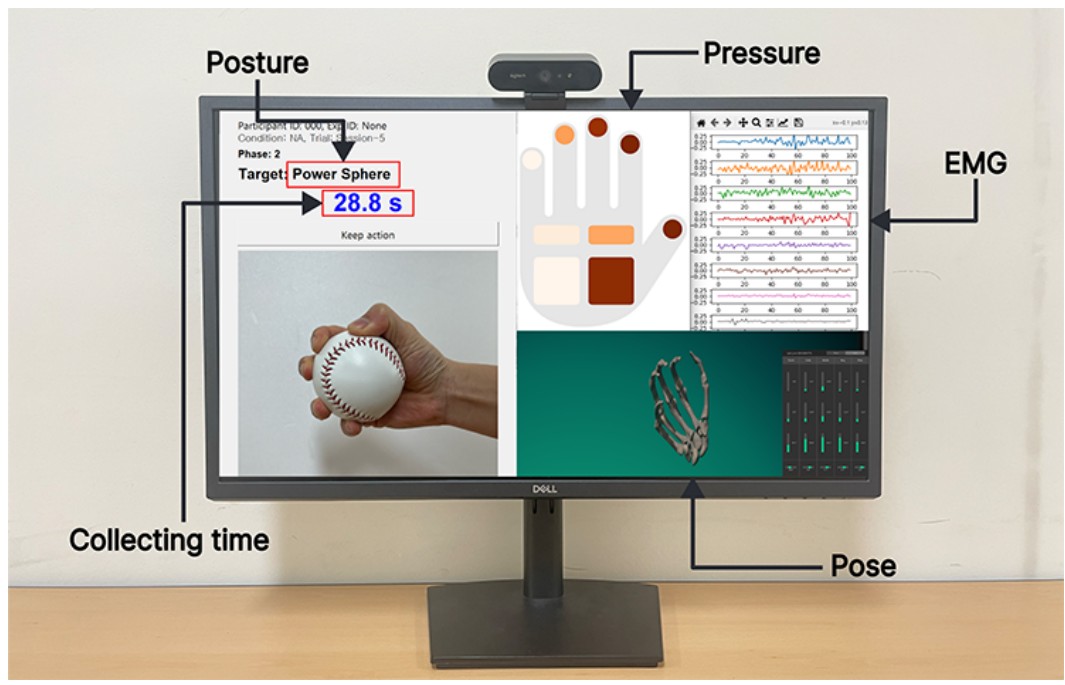

Figure 10: Data collection interface displaying real-time feedback for hand posture, pressure, and EMG signals, along with a timer for data acquisition sessions.

## B.4 Data Processing

### B.4.1 sEMG Signal and Pressure Data

With a frame length of 1,248 samples, corresponding to 0.624 seconds, we utilized a Short Time Fourier Transform (STFT) for sEMG signal. This method, leveraging a window length of 256 and a hop length of 32 samples, employed a Hamming window function to minimize spectral leakage. To preprocess the sEMG signal, we applied the STFT and considered only the signals below 64Hz to eliminate high-frequency noise, optimizing the signal's relevance for muscle activity analysis [4]. For the pressure data, considering the conventional force that a human can comfortably apply with their hand, we propose a maximum inferable force of 20 N, and forces exceeding 20 N were clipped to be treated as 20 N. Additionally, to exclude noise data caused by subtle sensor presses, values below 0.2 N were processed to be treated as 0.

### B.4.2 3D Hand Pose

For the 3D hand pose data, we use data obtained from the Quantum Mocap Metaglove for the training dataset. The hand skeleton model adopted in this study is as shown in Figure 11. The joints of the fingers from the middle to the ring finger are omitted because they are defined identically to those of the index finger. The raw data obtained through the data glove represents the hand as 20-dimensional angular data, denoting each finger with one abduction angle and three flexion angles [72]. For the thumb, it includes the CMC (carpometacarpal joint)'s abduction and flexion angles, along with the MCP (metacarpophalangeal joint) and IP (interphalangeal joint) flexion angles. For the other fingers, it encompasses the MCP's abduction and flexion angles, as well as the PIP (proximal interphalangeal joint) and DIP (distal interphalangeal joint) flexion angles. This 20-dimensional angular data is transformed into a three-dimensional angular

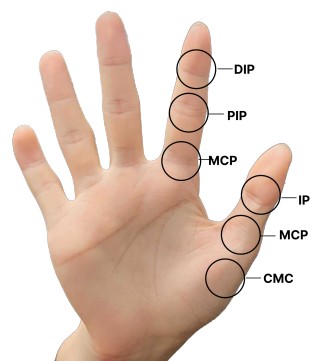

Figure 11: Hand skeleton model.

representation $\theta$ for each of the three joints determining the state of each finger, considering the kinematic tree of the hand skeleton in the MANO hand model [73]. Subsequently, through forward

Table 7: Relative bone length measurements for different finger sections.

| Hand Bone | Thumb | Index | Middle | Ring | Pinky |
|---|---|---|---|---|---|
| Root-to-MCP (Root-to-CMC for thumb) | 0.5134 | 1.0475 | 1.0000 | 0.9871 | 0.9585 |
| MCP-to-PIP (CMC-to-MCP for thumb) | 0.4225 | 0.4509 | 0.5375 | 0.5095 | 0.3392 |
| PIP-to-DIP (MCP-to-IP for thumb) | 0.3772 | 0.3014 | 0.3427 | 0.3039 | 0.2551 |
| DIP-to-Tip (IP-to-Tip for thumb) | 0.3324 | 0.2765 | 0.3061 | 0.2822 | 0.2540 |

kinematics [74], this is converted into a hand joint representation $J$ that includes the fingertips and the hand root. The relative bone lengths of each hand skeleton used during forward kinematics are shown in Table 7. When converting hand joint representation $J$ into 3D heatmap volumes $H$, $J$ is first min-max linearly scaled to take values in the range [12, 36], considering the maximum and minimum values for each spatial dimension in the training dataset. Thereafter, $J$ is transformed into 21 3D heatmaps $H$ by converting it into a 3D Gaussian heatmap with a standard deviation of $\sigma = 1$ for each dimension.

When using a 3D hand pose inferred from vision data, it can be transformed into the same representation as the 3D joint representation $J$. However, considering the global rotation of the wrist in the photo, it is necessary to align the direction of the global rotation of the hand with the kinematic tree of the hand used in training, and the scale of the hand also needs to match that used during training. Accordingly, the 3D hand pose $J$ inferred from vision data was rescaled so that the distance from the root joint to the MCP of the middle finger is identical, rotated so that the plane formed by the three joints (root joint, MCP of the middle finger, MCP of the index finger) is consistent, and translated so that the root joint is in the same position. Afterward, the 3D hand pose inferred from vision data is transformed into 21 3D heatmaps $H$ through the process described above.

## C  Detailed Description for Method

### C.1  Empirical Motivation of Our Framework

To empirically motivate our approach, we observed that sEMG signals alone might not be sufficient for fine-grained pressure localization on the hand. We conducted an empirical analysis demonstrating the limitations of using sEMG signals alone for this purpose. Figure 12 presents detailed sEMG signal patterns for two pairs of hand postures: (1) I-Press (index finger press) versus M-Press (middle finger press), and (2) TI-Pinch (thumb-index pinch) versus TM-Pinch (thumb-middle pinch). In each case, participants were instructed to apply pressure using different fingers, targeting specific regions of the hand. The 8-channel sEMG signals were recorded over a 5-second interval, with each channel representing muscle activity from different forearm muscles. However, as shown in Figure 12, the sEMG patterns are remarkably similar within each pair of actions, despite the differences in the fingers exerting pressure.

This similarity arises because the muscle activations required for pressing or pinching with adjacent fingers can produce overlapping sEMG signals due to the anatomical proximity and shared muscle groups involved. Consequently, relying solely on sEMG signals makes it challenging to distinguish between pressures applied by the index or middle finger. While sEMG signals provide valuable information about the force exerted, they lack the spatial specificity needed for precise pressure localization. These findings highlight the necessity of incorporating 3D hand posture information to disambiguate the source of pressure. By fusing sEMG signals with precise hand pose data, our framework can leverage both the force-related information from sEMG and the spatial context provided by hand posture, leading to more accurate and fine-grained pressure estimation.

### C.2  Details of Deep Neural Network Architecture

**sEMG Feature Extractor** $f_{\mathbf{EMG}}$    The sEMG feature extractor employs a conventional encoder-decoder architecture [4], with the encoder composed of three encoder blocks and the decoder comprising three decoder blocks. Each encoder block includes a 2D convolutional layer that filters the input, extracting pertinent features by convolving the input with a kernel of size (3, 3) and preserving the input size with padding set to *same*. 2D batch normalization and a ReLU activation function follow this. A max-pooling layer then downsamples the output by selecting the maximum value within

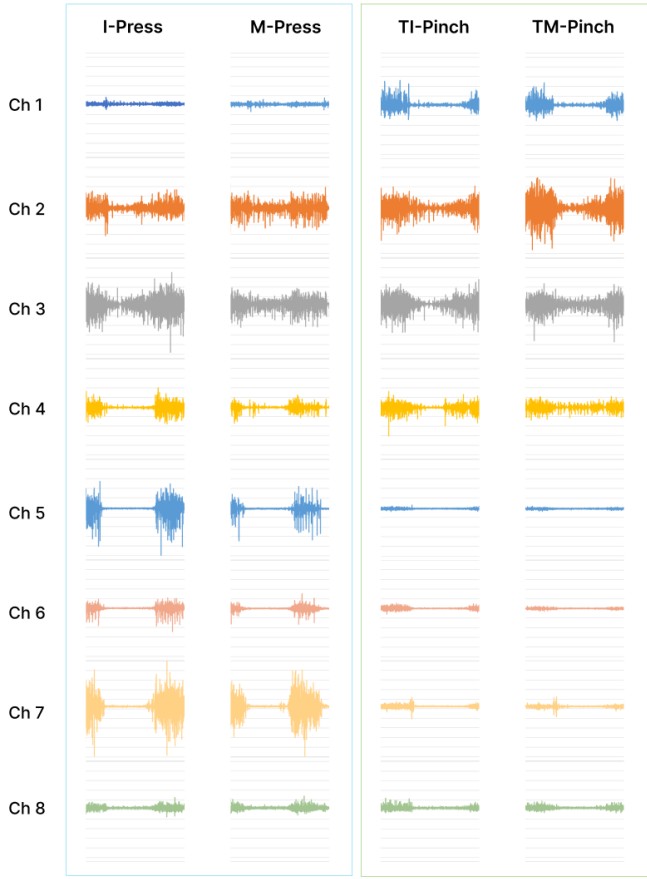

Figure 12: Visualization of patterns with similar EMG footprint on different postures.

each pooling window, as defined by the downsampling parameter, effectively diminishing the spatial dimensions. This block structure is iterated with increasing channel sizes and downsample rates. The initial encoder block processes the raw sEMG signals with 32 filters, followed by subsequent blocks that increase the filter count to 128 and 256.

In parallel to the encoder, each decoder block is designed to upscale the condensed features to a higher resolution. It utilizes a 2D convolutional layer with an identical (3, 3) kernel size and *same* padding, followed by batch normalization and ReLU activation, mirroring the encoder block. An upsampling step follows, augmenting the spatial dimensions to correspond to those in the encoder's earlier stages. The decoder restores the feature map from its condensed state, as generated by the encoder, to a higher resolution, employing decoder blocks in reverse sequence. Subsequently, the 2D feature map, having been processed through both the encoder and decoder, is flattened into a 1D feature vector. Finally, the 1D feature vector is projected to a 512-dimensional space through a single fully connected layer.

**Hand Pose Feature Extractor** $f_{hand}$   The Hand Pose Feature Extractor adopts a modified version of the ResNet34 architecture [75], incorporating 3D convolutions to adeptly handle spatial structure inherent in hand pose data [67, 76]. This adaptation is crucial for capturing the 3D pose of hand postures, offering a richer representation than traditional 2D models. Central to this architecture is the 3D ResNet block, termed the residual block. Each residual block starts with a 3D convolutional layer, employing a kernel size of $(3, 3, 3)$ and stride of 1, maintaining spatial dimensions through padding. This convolution is followed by 3D batch normalization and a ReLU activation, ensuring non-linearity and normalization of activations. A second 3D convolution with the same kernel size and padding repeats this process. The block is designed to accommodate optional downsampled

input, aligning input and output dimensions via a downsample path if necessary, employing either average pooling or a $(1, 1, 1)$ convolution for dimension matching, followed by batch normalization. Residual connections in each block allow inputs to bypass layers, helping to prevent the vanishing gradient problem by directly adding inputs to outputs.

The architecture initiates with a 3D convolution layer that processes the input using a kernel size of $(7, 7, 7)$, stride $(2, 2, 2)$, and padding to keep spatial dimensions consistent. It continues with four sequential layers comprising multiple residual blocks. These layers progressively increase channel depth from 64 to 512, adhering to the sequence $[64, 128, 256, 512]$. The layer structure is $[3, 4, 6, 3]$, indicating the number of residual blocks per layer. Downsampling occurs in the transition between these layers to reduce spatial dimensions and increase the receptive field. After progressing through the convolutional blocks, an adaptive average pooling layer condenses the 3D feature map to a size of $1 \times 1 \times 1$, effectively distilling the spatial-temporal features into a singular, compact feature vector. This vector is then connected to a fully connected layer, which outputs the 512-dimensional final feature representation.

**Our Full Model** $f$     After concatenating the 512-dimensional feature vectors from sEMG and 3D hand pose data, the model processes the resulting 1024-dimensional vector through a linear layer to reduce it to 256 dimensions. This reduction is immediately followed by a batch normalization step and a ReLU activation to standardize and introduce non-linearity into the features, enhancing their representation for the task at hand. This sequence—comprising a 256-dimensional linear layer, batch normalization, and ReLU activation—is executed twice to further refine the features. The refined features then advance through a Sigmoid-activated linear layer, tailored to the specific classification task with dimensions set by the numbers of interest regions ($I = 9$), finalizing the model's output preparation.

## C.3   Model Training and Inference

For the training of our model, we configured our setup as follows: We trained the model for 50 epochs to ensure adequate training steps. We set the batch size to 64 to balance computational efficiency and training stability. The model was optimized using the Adam optimizer [77] with a learning rate of 0.0001 and $(\beta_1, \beta_2)$ of (0.9, 0.999), along with a weight decay of 0.0001. The model training was conducted on a computing environment equipped with an AMD EPYC 7763 64-Core Processor, 1.0TB RAM, and an NVIDIA RTX 6000A 48Gi. Our model's entire training process took approximately 12 hours.

In our inference setup, we used a computer equipped with a 12th Generation Intel Core i9-12900K processor, 32GB of RAM, and an NVIDIA GeForce RTX 4080 with 16GB of GPU memory. The total number of parameters in our model amounts to approximately 66.17 M, with the sEMG feature extractor contributing 1.26 M parameters and the hand pose feature extractor contributing 64.11 M parameters. Our analysis on inference time indicated that our model requires only 4 milliseconds to process a single batch, demonstrating its capability to handle 256 streaming inputs in a batch inference setup without any delay or memory overflow issues on our test environment. Thus, we conclude that our model is sufficiently lightweight and efficient for real-time processing, making it practical for applications that require immediate feedback.

# D   Results

## D.1   Implementation Details of Comparative Methods

The model architecture and data processing details of the *sEMG Only Model* largely align with the official implementation[2] disclosed in [4]. One exception to this conformity is our modification of the final linear layer, where we expanded the number of prediction dimensions from 5 to 9. This alteration was made to estimate the force exerted at the tips of each finger, aligning the output dimensions with those of our study. Thus, this implementation serves as a counterpart solely utilizing EMG data, excluding hand pose information. For *PressureVision++*, we directly utilized the implementation and pre-trained weights officially released by the authors[3]. It is noteworthy that we initially considered

---

[2]https://github.com/NYU-ICL/xr-emg-force-interface
[3]https://github.com/pgrady3/pressurevision2

using PressureVision, which infers the entire hand's pressure from photos. However, after testing its official implementation[4], we found it entirely non-functional in our environment due to its lack of generalization ability. Consequently, we adopted PressureVision++, which is limited to estimating pressure at the fingertips but has a much higher ability for generalization.

The *3D Hand Posture Only Model* corresponds to our full model with the sEMG feature extractor $f_{\text{EMG}}$ removed. Consequently, the linear layer that previously mapped a 1024-dimensional concatenated feature to 256 dimensions has been adjusted to map from 512 dimensions to 256 dimensions. This modification reflects the absence of sEMG features, focusing solely on the utilization of 3D hand pose data. The *sEMG + Hand Angles Model* modifies our full model by replacing $f_{\text{hand}}$ with a 3-layer fully connected network, setting all layers to map features to a 256-vector. Each layer incorporates 1D batch normalization and ReLU activation, aiming for a consistent feature representation. Additionally, $\theta$ underwent dimension-wise min-max normalization to adjust values to the [0, 1] range, facilitating standardized input for model processing. For aspects not specifically mentioned, such as data processing, model architecture, and the choice of stochastic optimizer, we adhered to the configurations described for our full model. This approach ensures that our comparative analysis remains grounded in a consistent methodological framework, allowing for a fair evaluation of the different models' performance.

## D.2  Evaluation Metrics

We employ a set of evaluation metrics that allow for a comprehensive assessment of both classification and regression capabilities. These metrics are designed to quantify the accuracy, precision, and generalizability of our model in predicting pressure exertions across the hand. The metrics include:

**Coefficient of Determination ($R^2$).** The $R^2$ value measures the proportion of variance in the dependent variable that is predictable from the independent variables. It is defined as:

$$R^2 = 1 - \frac{\sum_{t=1}^{T} \sum_{i=1}^{I} (P_{i,t} - \hat{P}_{i,t})^2}{\sum_{t=1}^{T} \sum_{i=1}^{I} (P_{i,t} - \bar{P}_{i,t})^2}, \tag{3}$$

where $T$ is the number of time frames, $I$ is the number of hand regions, $P_{i,t}$ is the actual pressure, $\hat{P}_{i,t}$ is the predicted pressure for region $i$ at time $t$, and $\bar{P}_{i,t}$ represents a mean pressure for region $i$. This metric quantifies how well the model's predictions fit the actual pressure data, with values closer to 1 indicating a stronger correlation between predicted and actual pressures.

**Normalized Root Mean Squared Error (NRMSE).** NRMSE provides a normalized measure of the deviation of the predicted pressure values from the actual pressures applied, offering insight into the model's precision in estimating the magnitude of forces. It is computed as follows:

$$\text{NRMSE} = \frac{1}{P_{\text{max}}} \sqrt{\frac{1}{T \cdot I} \sum_{t=1}^{T} \sum_{i=1}^{I} (P_{i,t} - \hat{P}_{i,t})^2}, \tag{4}$$

where $P_{\text{max}}$ is the maximum observed pressure in the dataset.

**Classification Accuracy.** This metric assesses the model's capability to accurately classify whether pressure is being exerted by any region of the hand. Specifically, a prediction is considered correct only if the pressure exertion status of all hand regions (fingertips and palm areas) is accurately classified for a given time frame. This metric is crucial for understanding the model's ability to distinguish between active and inactive pressure application scenarios.

---

[4] https://github.com/facebookresearch/PressureVision

### D.3 Additional Quantitative Results

#### D.3.1 Mean Average Error

Table 8: Performance comparison of models in terms of MAE. Note that this table reports the performance of Table 2 in terms of the MAE metric.

| Method | MAE |
|---|---|
| sEMG Only [4] | $6.03 \pm 2.24$ |
| 3D Hand Posture Only | $9.03 \pm 4.18$ |
| sEMG + Hand Angles | $5.75 \pm 2.37$ |
| **PiMForce (Ours)** | **4.99** $\pm 2.02$ |

Table 9: Performance comparison of hand regions in terms of MAE. Note that this table reports the performance of Table 4 in terms of the MAE metric.

| Method | Finger Tip | | | | | | Hand Palm | | | | | Overall Mean |
|---|---|---|---|---|---|---|---|---|---|---|---|---|
| | Thumb | Index | Middle | Ring | Pinky | Mean | Upper Right | Upper Left | Lower Right | Lower Left | Mean | |
| sEMG Only [4] | 9.05 ± 4.19 | 8.40 ± 4.22 | 6.98 ± 3.62 | 5.10 ± 2.54 | 3.06 ± 1.08 | 6.52 ± 3.12 | 8.04 ± 4.46 | 7.40 ± 4.23 | 7.75 ± 4.30 | 6.73 ± 4.42 | 7.48 ± 4.35 | 6.95 ± 3.67 |
| 3D Hand Posture Only | 13.62 ± 7.00 | 13.30 ± 7.51 | 11.31 ± 6.65 | 8.07 ± 4.81 | 4.17 ± 1.55 | 10.09 ± 5.50 | 11.92 ± 7.44 | 9.88 ± 7.44 | 11.00 ± 7.29 | 9.03 ± 7.43 | 10.46 ± 7.40 | 10.26 ± 6.35 |
| sEMG + Hand Angles | 8.62 ± 4.08 | 8.04 ± 4.19 | 6.71 ± 3.63 | 4.87 ± 2.35 | 2.97 ± 1.01 | 6.24 ± 3.05 | 7.66 ± 4.33 | 6.34 ± 4.44 | 6.77 ± 4.56 | 5.75 ± 4.50 | 6.63 ± 4.46 | 6.42 ± 3.68 |
| **PiMForce (Ours)** | **7.38** ± 3.43 | **6.95** ± 3.44 | **5.91** ± 2.97 | **4.55** ± 2.36 | **2.61** ± 0.77 | **5.48** ± 2.59 | **6.56** ± 3.64 | **5.48** ± 3.72 | **6.07** ± 3.59 | **4.99** ± 3.76 | **5.77** ± 3.68 | **5.61** ± 3.08 |

#### D.3.2 Cross-User Evaluation and Comparison with PressureVision++

To assess the generalizability of our hand pressure estimation framework across different individuals, we conducted cross-user evaluations. Additionally, we performed a quantitative comparison with the vision-based method PressureVision++ to benchmark our model's performance against existing state-of-the-art approaches.

**Cross-User Evaluation** Given the known variability of sEMG signals due to anatomical differences and sensor placement among users, it is crucial to evaluate how our model performs when encountering data from participants not seen during training. We partitioned our dataset of 21 participants into a training set and a test set. Specifically, data from 17 participants were used to train the model, while data from the remaining 4 participants were reserved for testing. This ensures that the model is entirely blind to the test participants during both training and evaluation phases.

Based on the cross-user evaluation results presented in Table 3, our PiMForce combining sEMG signals with 3D hand posture data significantly outperforms the sEMG-only baseline across all evaluation metrics. Specifically, our method achieves an $R^2$ score of 70.06%, compared to 47.90% for the sEMG-only model, indicating a substantial improvement in the proportion of variance explained by the model. The NRMSE is reduced from 14.14% to 10.70%, and the MAE decreases from 12.24% to 8.54%, demonstrating enhanced precision in pressure estimation. Additionally, the classification accuracy increases from 57.40% to 72.01%, reflecting a superior ability to correctly identify the presence or absence of pressure across hand regions. Table 10 shows the detailed performance in terms of hand regions.

These improvements highlight the effectiveness of integrating 3D hand posture information with sEMG signals to capture both muscle activation patterns and spatial context, thereby enhancing the model's generalizability to unseen users. The reduced variability in performance metrics, as indicated by the lower standard deviations, also suggests that our method is more consistent across different individuals.

Table 10: Cross-user performance comparison of whole hand regions in terms of NRMSE.

| Method | Finger Tip | | | | | | Hand Palm | | | | | Overall Mean |
|---|---|---|---|---|---|---|---|---|---|---|---|---|
| | Thumb | Index | Middle | Ring | Pinky | Mean | Upper Right | Upper Left | Lower Right | Lower Left | Mean | |
| sEMG Only [4] | 14.26 ± 5.33% | 13.95 ± 5.83% | 12.22 ± 5.44% | 9.25 ± 3.49% | 6.51 ± 2.03% | 11.24 ± 4.42% | 12.75 ± 5.95% | 10.96 ± 6.16% | 12.07 ± 5.78% | 10.20 ± 6.21% | 11.50 ± 6.03% | 11.35 ± 5.14% |
| **PiMForce (Ours)** | **10.88** ± 3.90% | **10.84** ± 4.24% | **10.10** ± 4.41% | **8.28** ± 3.47% | **5.81** ± 1.72% | **9.18** ± 3.55% | **9.74** ± 4.40% | **8.24** ± 4.68% | **9.05** ± 4.43% | **7.60** ± 4.77% | **8.66** ± 4.57% | **8.95** ± 4.00% |

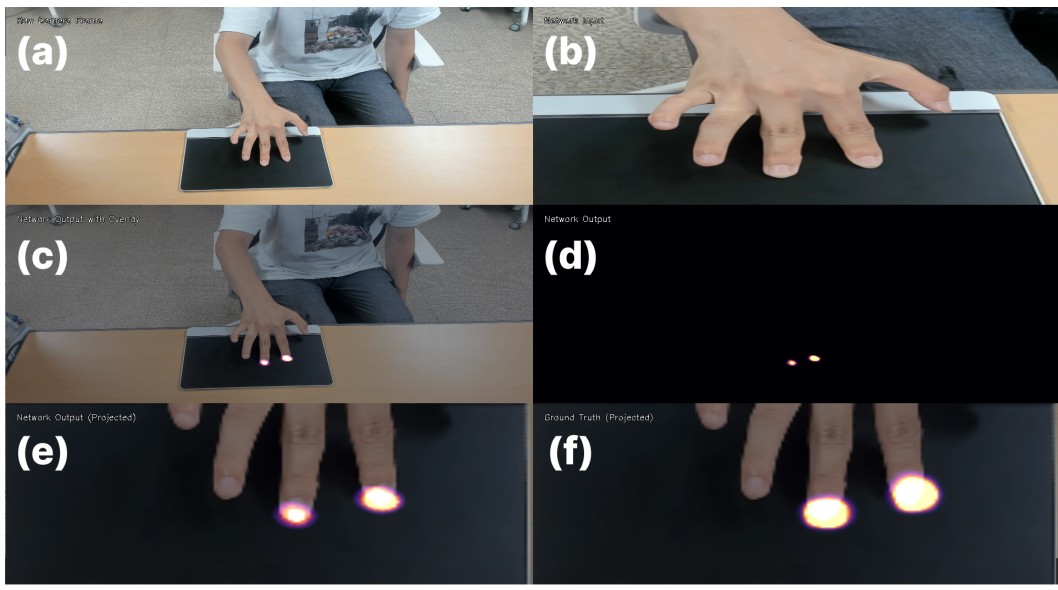

Figure 13: Visualization of ground truth pressure and predicted pressure for the same posture using the existing PressureVision++ hand pressure prediction framework. (a) Original image from camera. (b) Input image for the PressureVision++ model. (c) Overlaied predicted pressure by PressureVision++. (d) Original predicted pressure. (e) Overlaied predicted pressure by PressureVision++, projected onto Sensel pressure array. (f) Ground truth pressure, projected onto Sensel pressure array.

**Comparison with PressureVision++** To benchmark our PiMForce against existing vision-based pressure estimation methods, we conducted a quantitative evaluation of PressureVision++. Using the same equipment as the original PressureVision++ study — a Logitech Brio 4K webcam and a Sensel Morph pressure sensing array — we collected data from 5 participants. As PressureVision++ estimates pressure only on fingertips and requires full visibility of the hand within the camera view, we focused our evaluation on plane and pinch interaction sets, specifically: I-Press, M-Press, R-Press, P-Press, IM-Press, MR-Press, TI-Pinch, TM-Pinch, TIM-Pinch, TIMR-Pinch, and TIMRP-Pinch.

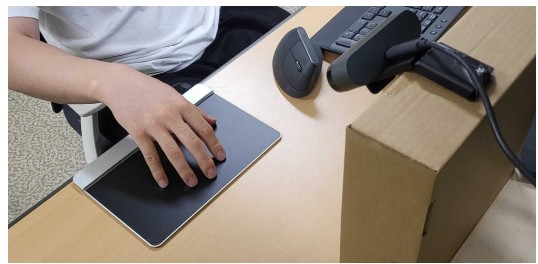

Figure 14: Collecting fingertip pressure data from a Sensel pad and hand data from an RGB camera to quantitatively evaluate PressureVision++ and our model.

Each participant was instructed to repeat each action for 30 seconds, following our data collection protocol. To ensure optimal visibility for PressureVision++, we carefully adjusted the camera angle to capture both the fingers and palm (see Figure 14). Note that the Sensel Morph could not measure thumb force during pinch actions, so this data was excluded from the analysis. Figure 13 illustrates ground truth pressure and predicted pressure under the setting of PressureVision++.

Table 5 presents the performance comparison between PressureVision++, our proposed method, and the sEMG-only model in the cross-user setting. It is important to note that our evaluation of PressureVision++ is inherently a cross-user performance report, as the model was trained on a separate dataset. As shown in the table, our method significantly outperforms both PressureVision++ and the sEMG-only model in all three metrics. Specifically, PiMForce achieves an accuracy of 82.20%, compared to 66.00% for PressureVision++ and 67.90% for the sEMG-only model. For a detailed breakdown of fingertip performance during plane and pinch interactions, we present the finger-specific performances in Tables 11. These results indicate that our proposed method consistently outperforms PressureVision++ in estimating fingertip pressures across all fingers during both plane and pinch interactions.

**Discussion** The cross-user evaluation confirms that our framework is robust and generalizes well to unseen participants. While there is a general decrease in performance when moving from within-user to cross-user evaluation (e.g., classification accuracy decreases from 83.17% to 72.01% for our method), our model still maintains superior performance compared to existing methods. PressureVision++ shows limitations in cross-user scenarios due to its reliance on visual features that can vary widely among individuals, such as skin texture, color, and hand shape. Additionally, it requires full visibility of the hand, limiting its applicability in scenarios where the hand is partially occluded or when interactions involve grasping objects.

Our PiMForce's superior performance can be attributed to the complementary nature of sEMG signals and 3D hand pose data. While sEMG provides insights into muscle activation patterns, 3D hand pose offers spatial context that helps disambiguate similar sEMG signals arising from different hand configurations. By integrating these modalities, our framework achieves more accurate and reliable hand pressure estimation across diverse users and interaction types, demonstrating its potential for practical applications in human-computer interaction and virtual reality systems.

Table 11: Cross-user performance comparison of finger tips in terms of NRMSE.

| Method | Plane | | | | | Pinch | | | | | Overall Mean |
|---|---|---|---|---|---|---|---|---|---|---|---|
| | Index | Middle | Ring | Pinky | Mean | Index | Middle | Ring | Pinky | Mean | |
| PressureVision++ [17] | 33.23 ± 1.55% | 29.14 ± 2.08% | 27.25 ± 1.67% | 15.48 ± 2.27% | 27.12 ± 1.37% | 43.18 ± 1.57 % | 42.46 ± 2.55 % | 31.72 ± 2.42 % | 27.77 ± 2.39 % | 36.93 ± 1.51 % | 32.79 ± 1.05% |
| sEMG only [4] | 13.90 ± 7.31% | 13.34 ± 8.18% | 8.72 ± 4.48% | 5.64 ± 1.93% | 10.40 ± 5.47% | 13.45 ± 7.15% | 11.12 ± 5.93% | 8.68 ± 4.81% | 5.15 ± 1.51% | 8.56 ± 5.92% | 10.34 ± 4.57% |
| PiMForce (Ours) | **10.13** ± 5.17% | **10.08** ± 5.40% | **8.49** ± 4.54% | **5.43** ± 2.42% | **8.54** ± 4.38% | **9.56** ± 4.79% | **8.48** ± 4.65% | **6.08** ± 2.81% | **4.58** ± 2.13% | **7.17** ± 3.59% | **8.34** ± 3.46% |

## D.4 Additional Qualitative Results

The video included in the supplementary material is designed to qualitatively demonstrate our model's inference performance in a time-continuous context.[5] The extended results of Section 5.2.2 and Figure 4a in the main text are displayed. Figures 15-18 serve as extended figures to Section 5.2.2 in the main text. Figures 19-22 act as extended figures to Figure 4a in the main text.

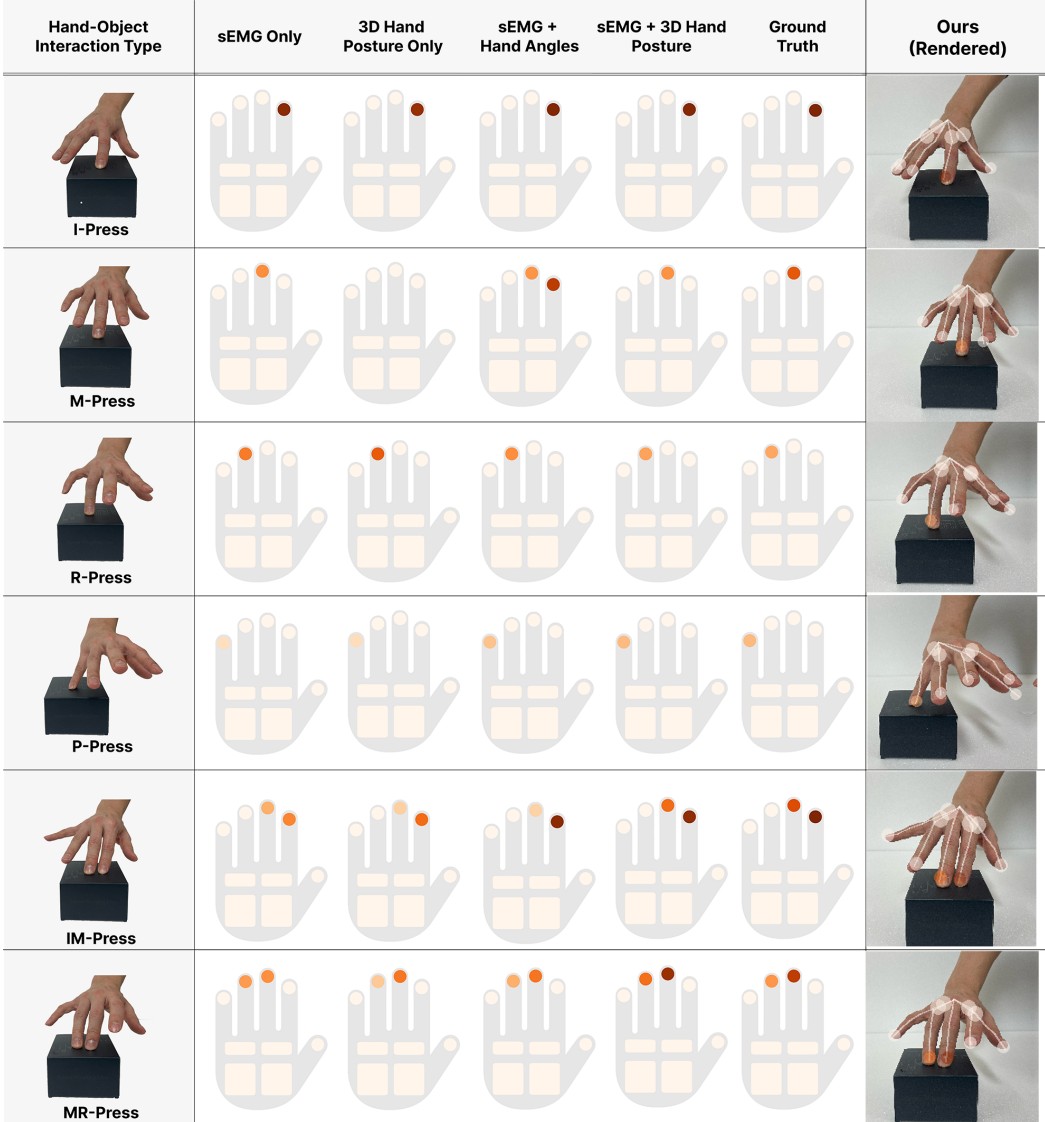

Figure 15: Qualitative visual analysis among comparative models.

---

[5] https://hci-tech-lab.github.io/PiMForce/

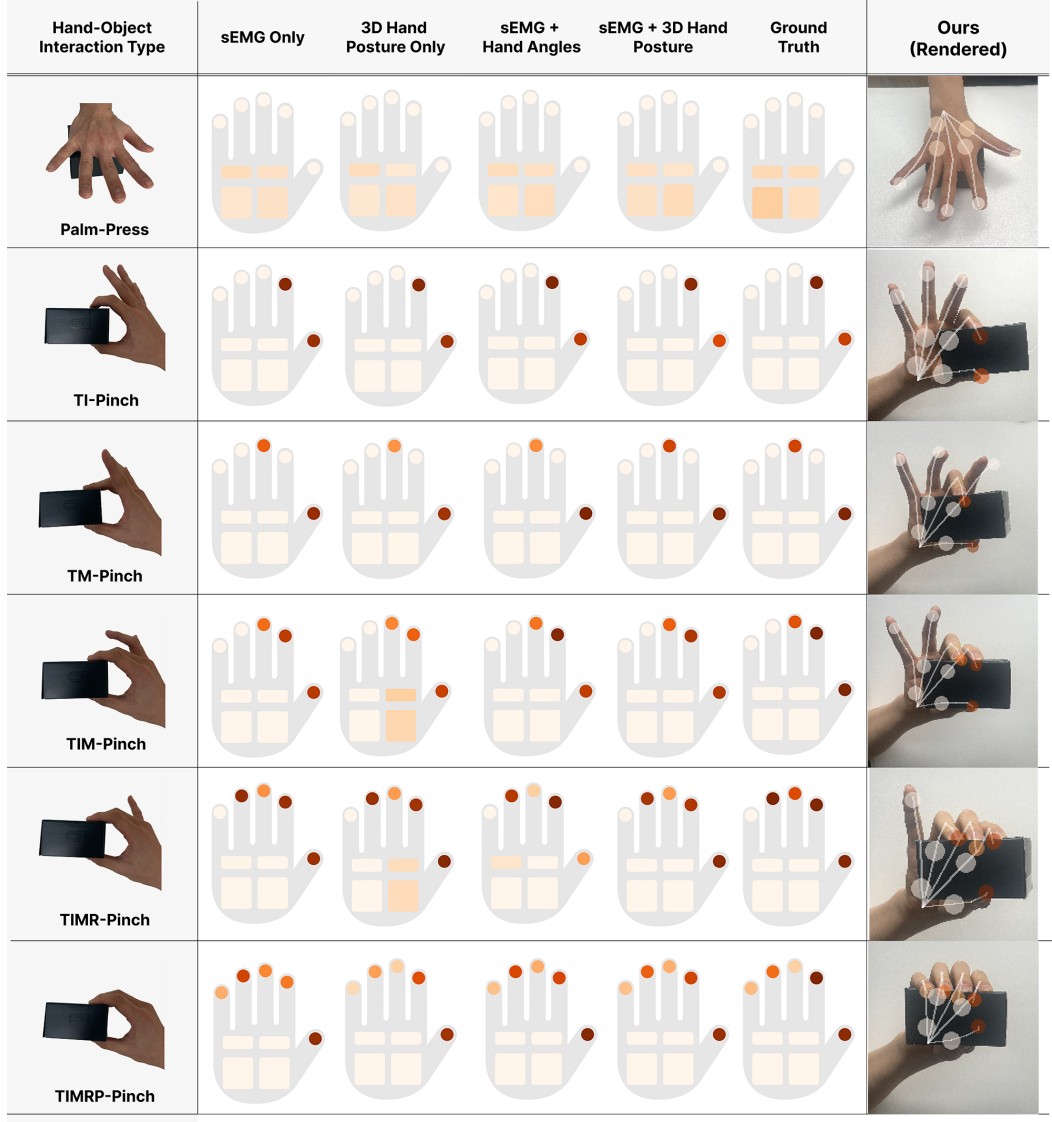

Figure 16: Qualitative visual analysis among comparative models.

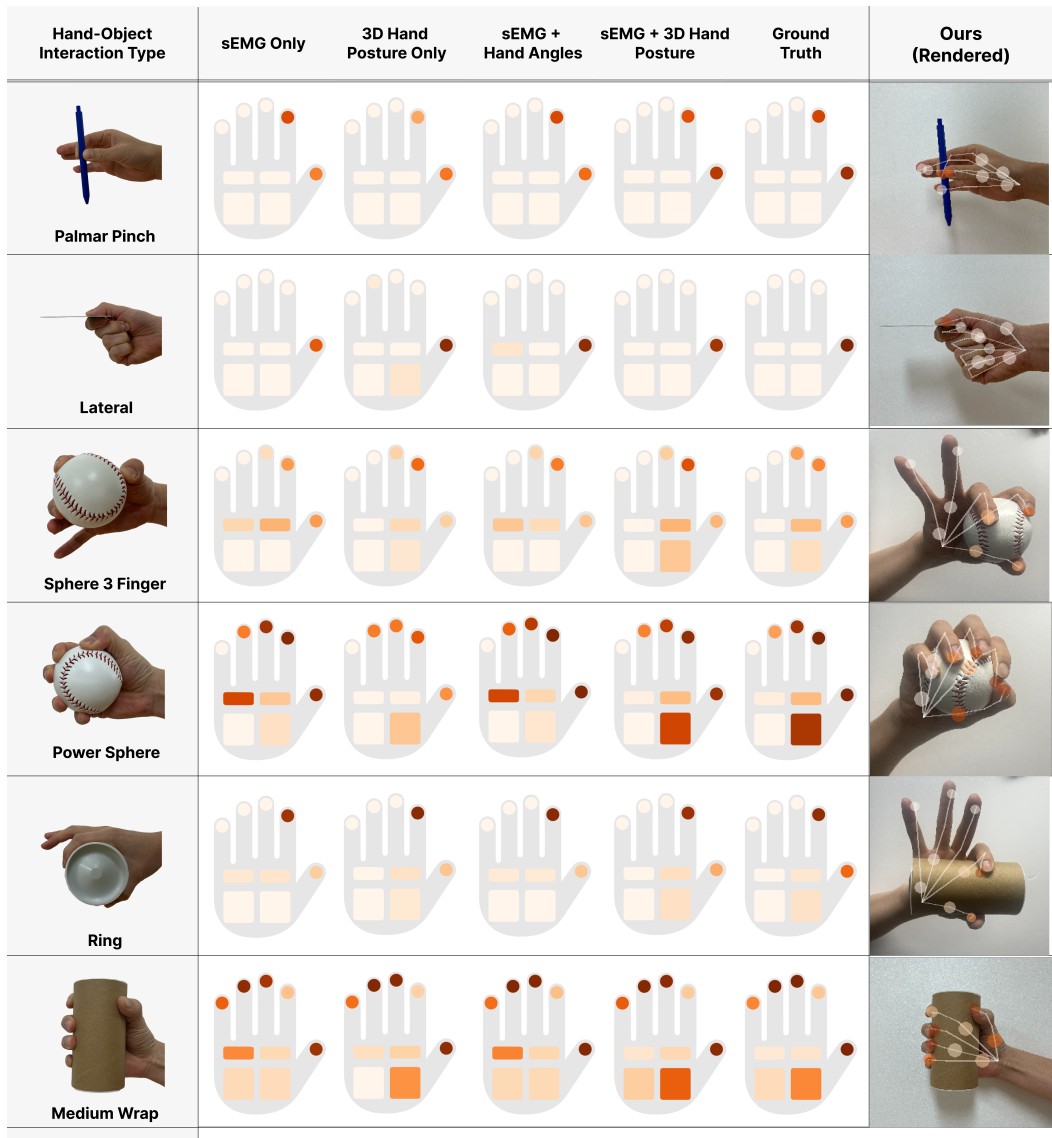

Figure 17: Qualitative visual analysis among comparative models.

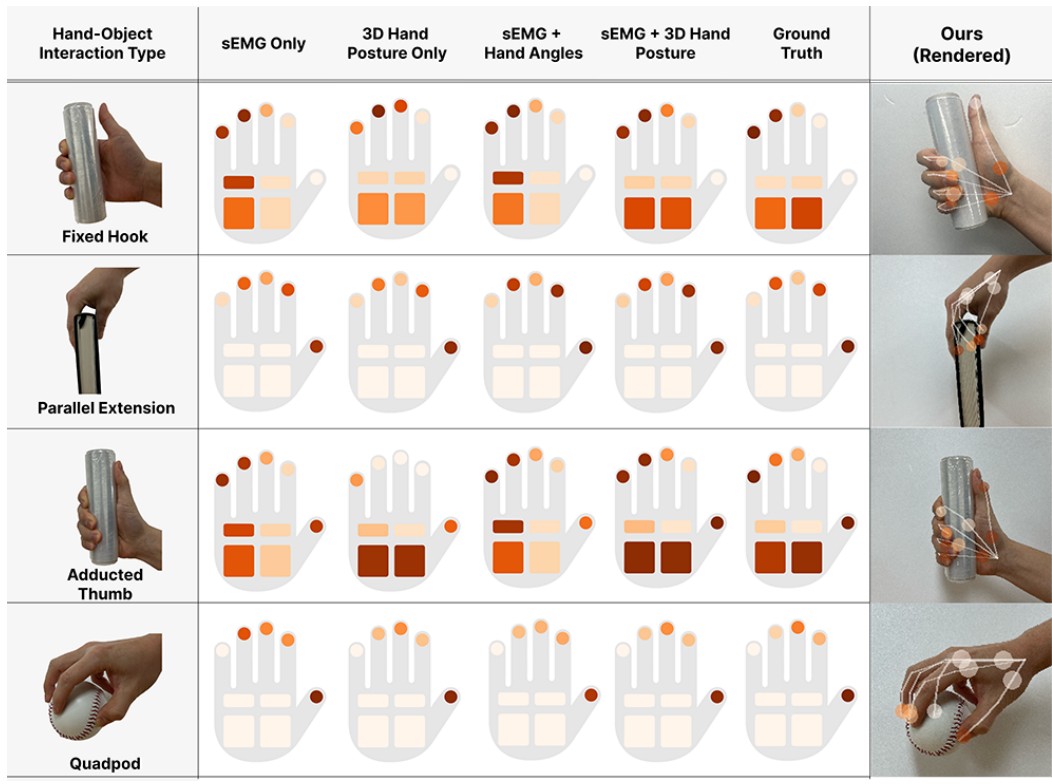

Figure 18: Qualitative visual analysis among comparative models.

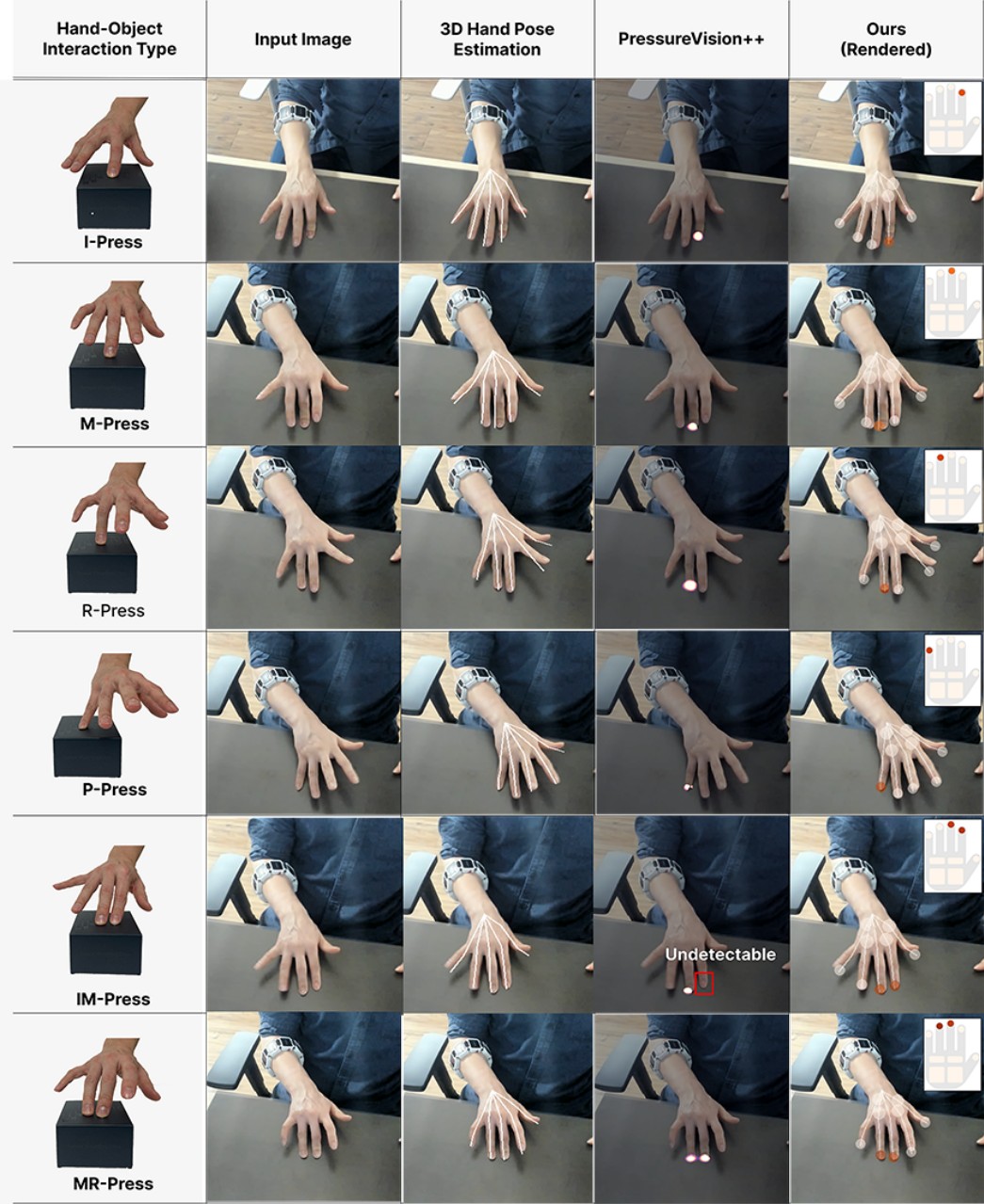

Figure 19: Pressure estimation results for the vision-aided hand in the test set.

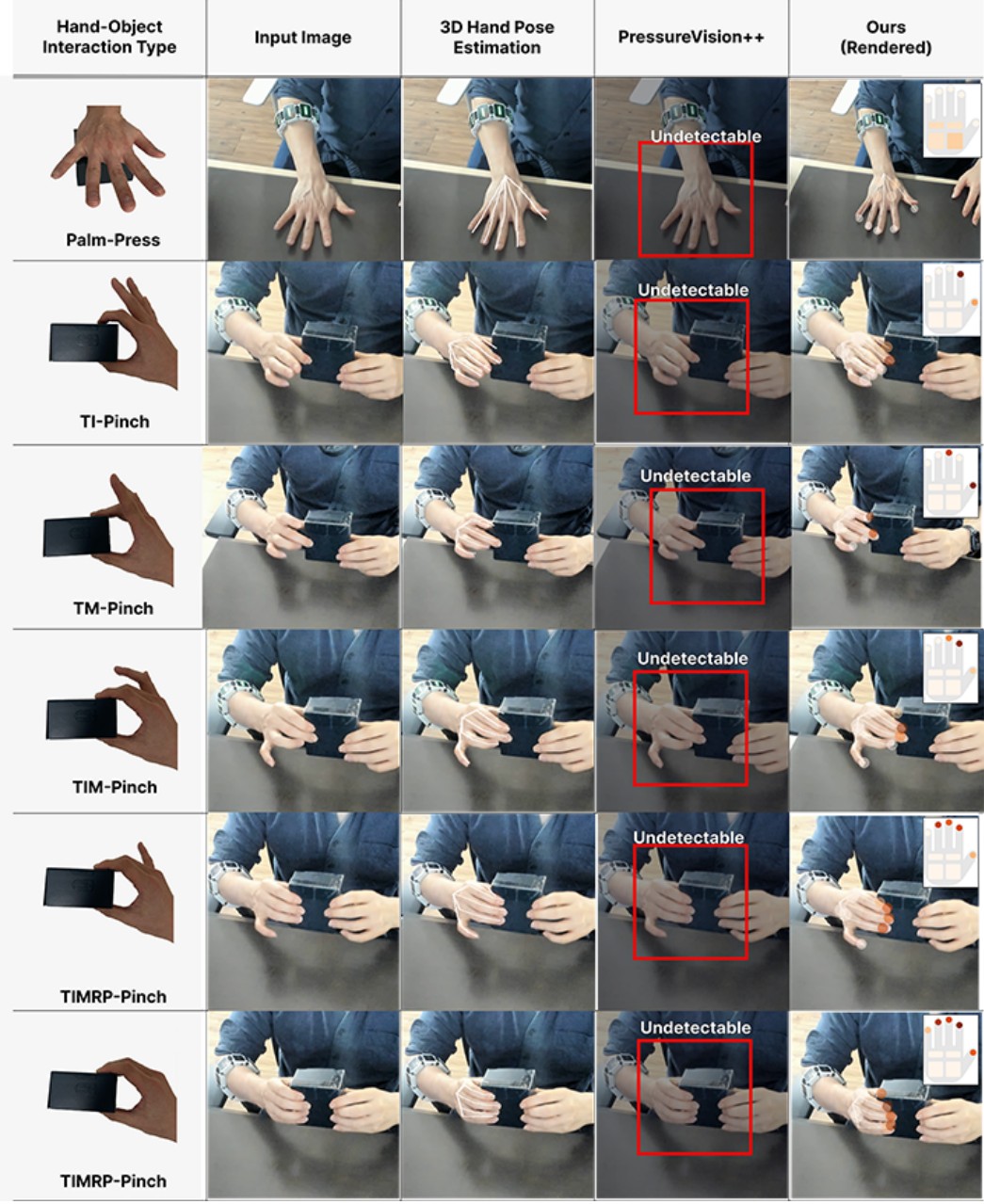

Figure 20: Pressure estimation results for the vision-aided hand in the test set.

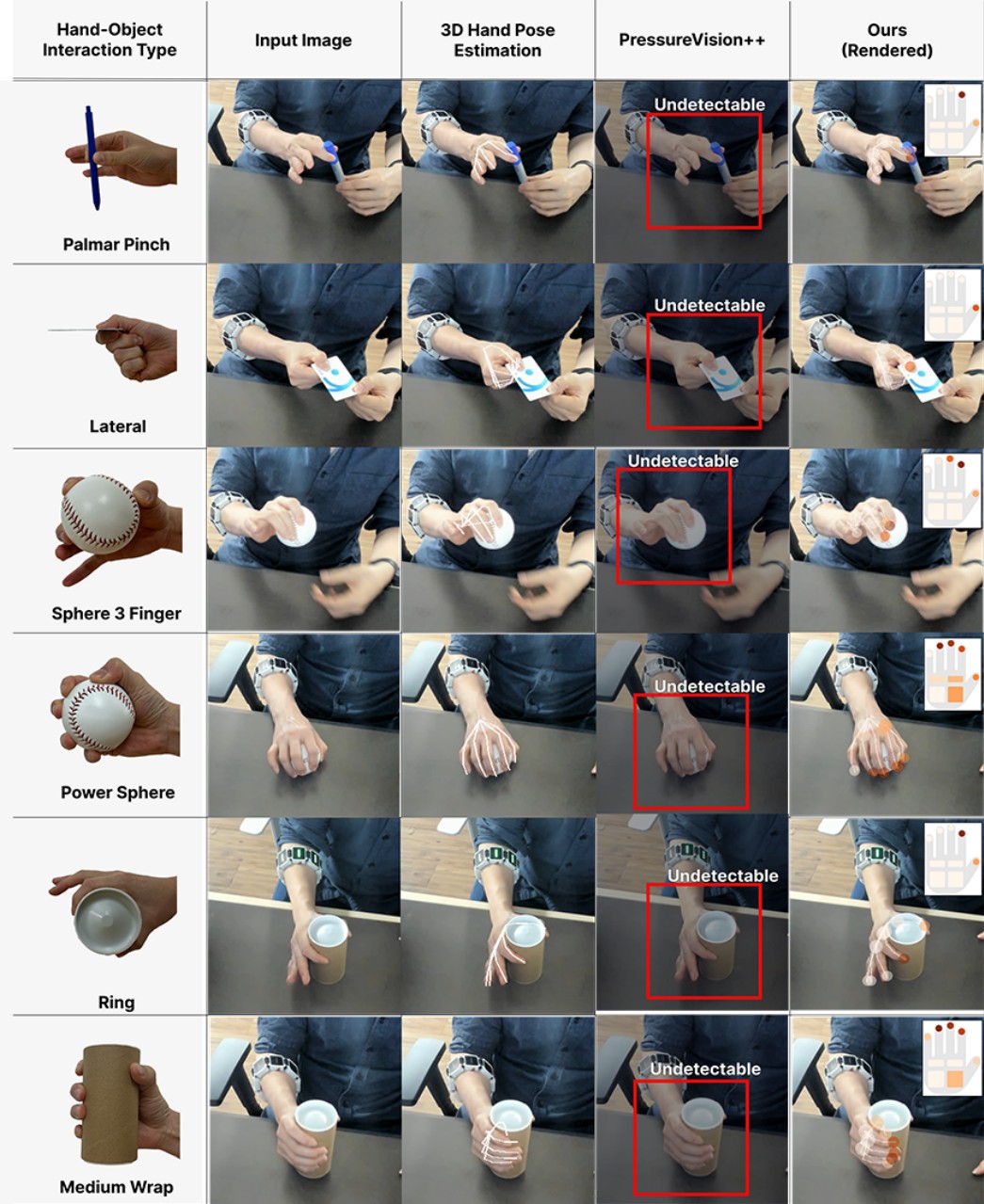

Figure 21: Pressure estimation results for the vision-aided hand in the test set.

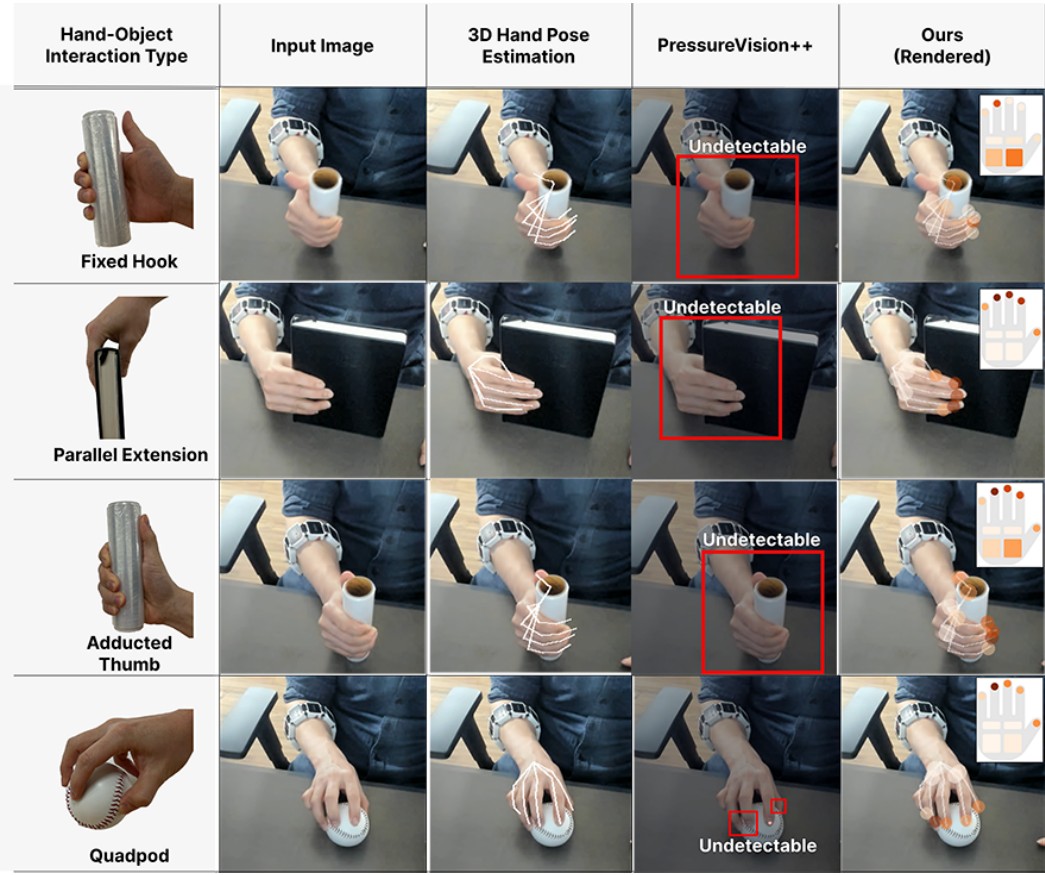

Figure 22: Pressure estimation results for the vision-aided hand in the test set.

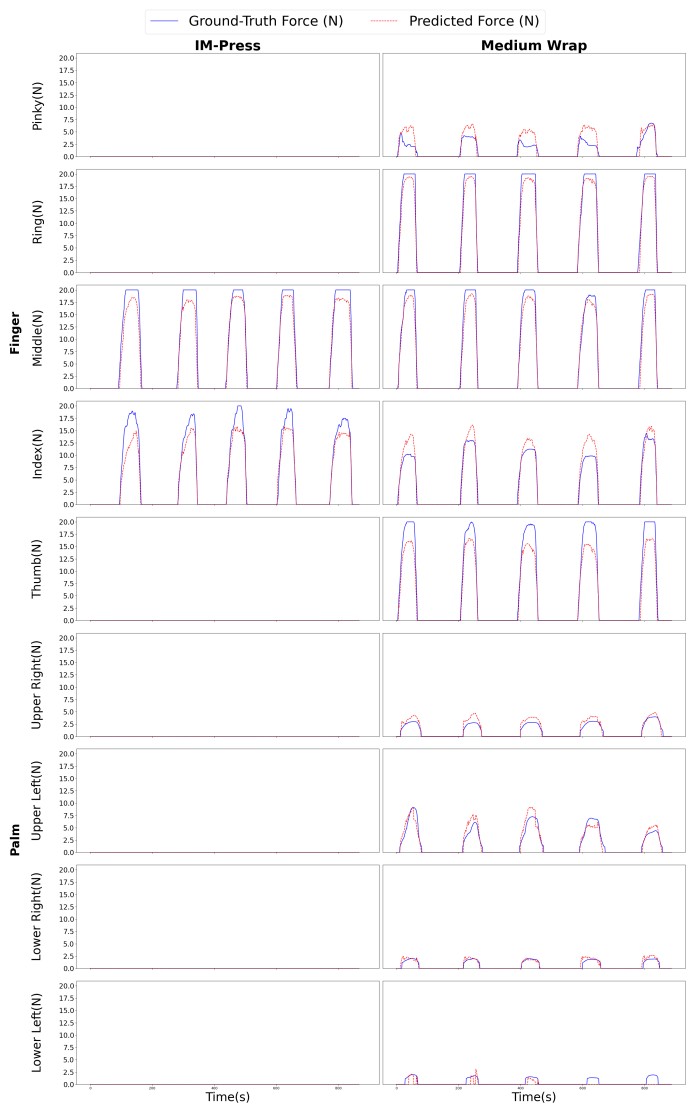

Figure 23: Comparison of fingertip and palm forces predicted by our model with ground truth data over time. To evaluate our model, we use the IM-Press posture, which uses only fingertip pressure, and the Medium Wrap posture, which uses whole-hand pressure, during Subject 5's test session.

## D.5    Comparison between Predicted Pressures and Ground Truth over Time

To provide a more detailed analysis of our model's temporal performance, we include additional figures that showcase the ground truth and predicted pressure values over time for various hand actions. Figure 23 displays the pressure trajectories for all nine hand regions during the execution of the TM-Press and Medium Wrap actions by a participant. In this 20-second interval, the participant performs the TM-Press action for the first 10 seconds, followed by the Medium Wrap action for the next 10 seconds.

In these plots, each subplot corresponds to a specific hand region, with time on the horizontal axis and pressure magnitude on the vertical axis. The solid lines denote the ground truth pressures measured by the tactile glove, while the dashed lines represent the pressures predicted by our model. The alignment between the ground truth and predicted pressures across different hand regions and actions demonstrates the model's capability to accurately track pressure dynamics over time. Notably, the pressure shifts from regions associated with the TM-Press action to those corresponding to the Medium Wrap action, reflecting the participant's change in hand posture and interaction.

# E   Limitations and Future Works

Our framework and model, while innovative, is not without its limitations. One of the primary constraints is its reliance on the accuracy of off-the-shelf hand pose detectors during inference. If the hand pose is inaccurately estimated, it can lead to erroneous pressure estimations. Furthermore, our analysis, as evidenced by Figure 6 of the main text, reveals that performance is not uniformly high across all hand postures. This inconsistency indicates areas that could benefit from further refinement. Due to the time constraints of participants, our dataset does not cover the full range of hand postures described in existing literature on hand posture taxonomy [55, 56]. Additionally, the absence of mesh information for both hand and object means we lack precise data on the surface contact locations. Our study also faces limitations due to the demographic homogeneity of the participants. The lack of diverse demographic representation among participants may limit the generalizability of our current model across different populations.

Overcoming these limitations opens up several directions for future work. Exploring additional modalities such as depth sensor, constructing an end-to-end model that integrates vision sensors with sEMG signals, adopting high-density electromyography (HD-EMG) equipment [78, 48] to surpass the limitations of standard sEMG devices, and expanding our dataset through more extensive participant sampling and long-term data collection are promising paths forward. Incorporating hand mesh and object mesh estimation [79–81] with our muscular force learning framework could provide more accurate information regarding contact points and the corresponding pressure on hand mesh. Demographic limitations can be addressed by conducting studies with a larger and more diverse participant pool. Expanding the participant demographics will enhance the generalizability of our results and ensure the framework's applicability across a broader range of users. These steps will help refine our model, making it more robust and widely applicable across different hand interactions.

Additionally, our current framework primarily leverages global features from both sEMG signals and 3D hand pose data, without fully exploring alternative multimodal fusion methods. We did not incorporate fine-grained spatial information or prior structural knowledge of hand poses into the model architecture. Future work could investigate different fusion techniques, such as integrating spatial hierarchies or attention mechanisms, to capture the intricate relationships between hand pose and muscle activity. This exploration could further enhance pressure estimation accuracy and robustness.

# F   Potential Social and Broader Impacts

The proposed framework has several promising applications across various fields. In healthcare and rehabilitation, this technology could enhance therapeutic devices by providing detailed feedback on hand movements and pressure, improving the effectiveness of rehabilitation exercises and assistive devices for individuals with motor impairments. In human-computer interaction, integrating hand pressure data could lead to more intuitive and responsive interfaces, particularly in virtual reality (VR) and augmented reality (AR) environments. In occupational safety and ergonomics, our framework could help design ergonomic tools and workspaces, reducing the risk of repetitive strain injuries and enhancing worker productivity and safety.

Despite its potential usefulness, the use of forearm-worn sEMG to estimate hand movements, poses, and pressure could introduce biases in the model's performance due to medical and physiological variations. For instance, some individuals have less muscles than others; for example, the palmaris longus muscle, which flexes the wrist joint, is absent in about 15% of the population [82]. To mitigate these risks, it is essential to consider a broader clinical demographic when recruiting participants and designing research for future studies.

While our study did not present any direct clinical risks to the participants, we recognize the importance of ethical data handling, particularly regarding sEMG signals which may contain personal information. Prior to participation, all participants were provided with a detailed information sheet outlining the purpose, procedures, and potential risks of the study. We obtained written informed consent from each participant, ensuring they understood the nature of the data being collected and their right to withdraw at any time. All collected data was anonymized, complying with relevant data privacy regulations

