# OpenReview forum: "Posture-Informed Muscular Force Learning for Robust Hand Pressure Estimation"
_NeurIPS.cc/2024/Conference — NeurIPS 2024 poster_

### Official Review · Reviewer_saS6 · 2024-06-20

**Soundness:** 3
**Presentation:** 3
**Contribution:** 3
**Rating:** 6
**Confidence:** 3

**Summary:**

The authors leverage 3D hand pose and sEMG signals as inputs to facilitate hand pressure estimation. With data gloves, a multimodal hand-object interaction collection system is devised. Empirical experiment results show the efficacy of the proposed method. Furthermore, the feasibility and robustness is proven in a single-camera scenario.

**Strengths:**

- The collected dataset is superior in scale and provides a holistic collection for various hand-object interactions.

- The ablation studies are helpful in providing useful insights.

**Weaknesses:**

- A highly related dataset ''ActionSense: A Multimodal Dataset and Recording Framework for Human Activities Using Wearable Sensors in a Kitchen Environment'' in NeurIPS 2022 should be cited and discussed.

- The evaluation is only conducted in a drop-one-session manner. It might be better if a drop-one-subject evaluation could be provided to demonstrate the cross-subject ability for a more general application.

- The results are impressive in their low relative errors. It would help more with the absolute errors reported.

- The results are reported with little temporal information. A curve showing the predicted force changing with time would be preferable. Also, quantitative metrics on the temporal consistency would be a bonus if possible.

- For glove-less evaluation, only qualitative results are reported. It appears possible to provide a quantitative comparison given the controlled hand postures. For example, attaching sensors to the contact region of the MR-Press makes it possible to provide GT in glove-less scenarios.

- Reporting quantitative prediction errors for different postures is a missing chance to provide more insights.

**Questions:**

- Are the readings from 6 sensors belonging to Region 8 simply added up as the pressure of Region 8?

- Is it possible to briefly explain how the quality would be influenced by muscle fatigue as mentioned in L158?

- Will the data glove introduce different interaction patterns from those without data gloves?

**Limitations:**

Please refer to the Weaknesses and Questions part.

---

> ### Author Rebuttal · Authors · 2024-08-07
>
> **(W1) Additional dataset reference**
>
> As you suggest, we will cite and discuss in related work section as follows:
> A highly relevant work is the ActionSense dataset, which focuses on capturing multimodal data of human activities in a kitchen environment using wearable sensors. Similar to our approach, ActionSense emphasizes the importance of rich, synchronized, multi-modal data streams for understanding human actions and gathers . However, while ActionSense provides a valuable resource for understanding general kitchen activities, our work focuses on utilization of 3D hand posture into muscular force learning for understanding hand pressure estimation.
>
> **(W2) Additional cross-user evaluation**
>
> Please refer (Joint Response 2).
>
> **(W3) Additional results regarding absolute errors**
>
> We have included the Mean Absolute Error (MAE) results in the rebuttal PDF file. Table R6 and Table R7 in the rebuttal PDF file correspond to Table 2 and Table 3 in the main manuscript, respectively, and present the MAE metrics for each comparison.
>
> **(W4) Providing temporal information for predicted forces**
> - **“A curve showing the predicted force changing with time would be preferable.”**: We acknowledge the reviewer's insightful suggestion regarding the need for temporal information in our results.  To illustrate the performance of our model over time, we will include figures showing the temporal evolution of both ground truth and predicted pressure values. Figure R1 in the rebuttal PDF file provides an example of such visualization. The figure presents the ground truth pressure and the pressure predicted by our model for all 9 hand regions during TM-Press and Medium Wrap actions. The horizontal axis represents time, each row corresponds to a specific hand region, and the vertical axis in each plot shows the pressure values. In this example, the user performs the TM-Press action for 10 seconds, followed by the Medium Wrap action for another 10 seconds.  As evident from the figure, the location of pressure exertion and prediction shifts according to the action performed. We believe that these temporal visualizations will enhance the readers' understanding of our framework by clearly demonstrating how data is collected and predictions are made over time. We will include these figures and the corresponding discussion in a new Section D.4 titled "Comparison between Predicted Pressures and Ground Truth along with time" in the revised manuscript. Moreover, we’ll add one of those figures in the main body of the paper as in (Line 299) of Section 5.2.1.
> - **“Quantitative metrics on temporal consistency would be a bonus if possible.”**: We would appreciate it if the reviewer could provide specific examples or references for "quantitative metrics on temporal consistency". This would allow us to better understand their suggestion and provide a relevant analysis.
>
> **(W5) Additional experiments and results for glove-less evaluation**
>
> Please refer (Joint Response 1).
>
> **(W6) quantitative prediction errors for different postures**
>
> You correctly pointed out that our study does not demonstrate the performance of our framework on unseen gestures during the training phase. We acknowledge this limitation. While constructing our dataset, we strived to collect data encompassing a wide range of hand interactions, including plane interactions, pinch interactions, and grasp interactions, to facilitate comparisons with previous works [R1,R3]. However, we also had to consider the limited time and potential fatigue of our participants. Therefore, we made a selection of representative postures, as shown in Figure 8 and Figure 9. Collecting additional data for novel gestures with new participants is infeasible in this rebuttal phase, preventing us from conducting further experiments on generalization. However, as highlighted in Lines 908 and 152, our selection of postures covers a considerable portion of representative movements from the reference grasp taxonomy and those presented in [R3]. Further research with new gestures and objects is certainly needed to solidify these points. Following this point, we will add this text into the limitation section:
> - While our dataset covers a range of representative hand postures, it is limited in its exploration of novel gestures and object interactions unseen during training. Further research is necessary to investigate the generalizability of our framework to new objects and gestures. Expanding the dataset with a broader range of hand-object interaction scenarios and evaluating the model's performance on these unseen examples would provide valuable insights into its robustness and adaptability.
>
> **(Q3)Will the data glove introduce different interaction patterns from those without data gloves?**
>
> As outlined in Section 4.4 of the main text, we describe how we canonicalize 3D hand pose data derived from a vision-based hand pose estimator in Section B.4.2, 3D Hand Pose. During inference, we align the 3D hand pose estimated from visual data with the training data representation. This process involves rescaling the hand pose to match the bone lengths used during training, rotating it to align with the kinematic tree of the MANO hand model, and translating it to ensure consistent root joint positioning. This canonicalization minimizes the discrepancy between hand pose data from the glove and the vision-based estimator, improving the generalizability of our model during inference. Therefore, if the inference of hand pose is good enough, there will be no difference between wearing gloves and not. As noted in the Section E. Limitations and Future Works in the appendix, our framework inherently relies on the accuracy of off-the-shelf hand pose detectors during inference (Line 1097). This dependency is acknowledged as a limitation of the proposed modality and inference pipeline.
>
> Due to limitations on the length of our rebuttal, we would like to address (Q1) and (Q2) in seperated comment.

---

> ### Author Response · Authors · 2024-08-07
> **Additional comment for Q1 and Q2**
>
> **(Q1) Are the readings from 6 sensors belonging to Region 8 simply added up as the pressure of Region 8?**
>
> Instead of summing the values from the 6 sensors within Region 8, we define the maximum value among those sensors as the pressure for that region. Consequently, individuals with smaller hands might not activate all 6 pressure sensors in Region 8 during certain grasp postures. Therefore, we opted for the maximum value among the 6 sensors rather than the sum. Similarly, we chose the maximum over the average for the same reason. When only a subset of the 6 sensors are activated due to hand size discrepancies, using the average would result in an artificially low pressure reading due to the inactive sensors.
>
> **(Q2) Is it possible to briefly explain how the quality would be influenced by muscle fatigue as mentioned in L158?**
>
> Muscle fatigue is the decline in a muscle's ability to generate force. It happens when a muscle is repeatedly contracted or held in a sustained contraction for an extended period. Imagine holding a heavy book at arm's length – after a while, your arm will start to shake, and it will become increasingly difficult to keep the book up. This is muscle fatigue.
> The connection between muscle fatigue and EMG-based control is that muscle fatigue can negatively impact the quality of EMG data used for hand pressure estimation. As muscles tire, their electrical activity changes. This can manifest as increased signal amplitude (RMS) and shifted frequency content[R4]. These changes make it harder for the model to accurately distinguish between different hand postures and pressure levels. Moreover, fatigue leads to less consistent muscle activation patterns, making it challenging for the model to learn reliable relationships between EMG signals and exerted pressure[R5].
> While muscle fatigue is recognized as an important topic in EMG-based control, it falls outside the scope of this study. To minimize its effects on our results, we ensured sufficient rest periods between data collection trials for both training and testing. To convey this consideration more explicitly, we will include a statement regarding this after Line 930 in Section B.3 Data Collection Protocol.

---

> > ### Comment · Reviewer_saS6 · 2024-08-12
> >
> > Thanks for the thorough responses. Most of my concerns are addressed.

---

> > > ### Author Response · Authors · 2024-08-12
> > >
> > > We again express our sincere gratitude to the reviewer for their insightful comments and constructive feedback. We appreciate the time and effort invested in carefully reviewing our work. The reviewer's suggestions have significantly contributed to enhancing the clarity, depth, and overall quality of our manuscript. We have diligently addressed each concern raised, and we believe the revisions made have strengthened our paper considerably. We are particularly grateful for the valuable guidance on presenting temporal information. We believe that the revised manuscript, incorporating the reviewer's feedback, offers a stronger contribution to the field and we are confident that it will be well-received by the NeurIPS community.

---

### Official Review · Reviewer_YYDP · 2024-07-09

**Soundness:** 3
**Presentation:** 3
**Contribution:** 2
**Rating:** 5
**Confidence:** 4

**Summary:**

This manuscript introduces a framework for estimating the pressure of 9 points on the hand during 22 distinct hand object interaction types including pinches, grasps and planar interactions. They use an 8 channel sEMG band placed at the forearm as well as hand pose estimated using a glove or monocular video camera. They train models using a dataset from 20 participant that uses pressure sensors in the glove as ground truth. They evaluate models within participants, and compare pressure estimation models that use different combinations of sEMG and hand pose, including different representations of hand pose using joint angles or embedded 3D coordinates. The best performance is achieved by a multimodal network that fuses an encoding of the spectrogram of the sEMG and the 3D hand pose, the later using a 3D convolutional network. They compare performance across fingers and gestures, and give qualitative comparisons to other vision-based approaches.

**Strengths:**

- The combination of different modalities integrated in the baseline is I believe new, and the method is evaluated across a sufficient number of participants (20) and gestures.
- They compare across relevant single-modality baselines, and break out evaluations broken out per-behavior.
- The framework is clearly defined and contextualized. I think the 3D pose embedding and joint angle comparison is nice.

**Weaknesses:**

- All evaluation is within-participant. sEMG is known to be quite variable across participants due to variations in anatomy and placement, and the hierarchy of performance across modalities will likely change.
- The comparisons with PressureVision++ (Figure 4) and other SOTA approaches are a little underwhelming, as they essentially show the framework only functions when all fingertips are tracked. A more informative set of comparisons would show results, with ground truth, for cases when the fingertips are unoccluded. For figure 16, the quantitative comparison betwene the two would be informative.
- The behavioral generalization to new objects and gestures is unclear. Similarly the performance with raw vision data is only qualitative. It is unclear how the system handles domain mismatches between the glove and vision-based hand pose estimation framework. In combination with the lack of cross-participant information the in the wild utility of the approach is unclear.

**Questions:**

** Questions **
- what is the cross participant performance?
- What is the quantitative performance compared to Pressure Vision? What is the correlation between the two techniques if an absolute ground truth (e.g. from sensel) is not possible.
- Why does utilizing hand angles not improve the model performance?
- How was 3D hand pose acquired in cases without the glove? I don't see a description of this method.
- Can you show raw traces of the discretized pressure to give an intuition for what the real pressure data looks like?

**Minor**

- **Datasets for Hand Pressure Estimation** — do these datasets include pressure, many of these appear to be object interaction papers
- Fig 4: can you clarify if this participant was in the training dataset?
- L302 ‘significant’ - were significance tests run here?
- Table 3 - what are the relevant error bars for this table?
- nit: L 336 - pionering ?
- Table 2:unclear what errors are over
- Table 3: unclear what value is being plotted
- all exocentric?
- nit: Figure 3 pixelated
- Manuscript overstates at multiple points ‘L316 ‘exceptional’

**Limitations:**

Yes, they review this in section F.

---

> ### Author Rebuttal · Authors · 2024-08-07
>
> **(W1) Additional experiments for cross-user evaluation**
>
> Please refer (Joint Response 2).
>
> **(W2-1) Clarification on the comparison with PressureVision++**
> You are correct that the current comparison might not be entirely fair, and we acknowledge that our framework currently performs best when all fingertips are tracked. However, we'd like to clarify our approach in two ways. First, this limitation is inherent to the PressureVision and PressureVision++ methods themselves. As you pointed out, PressureVision++ relies on visual cues like color and shape changes on the hand caused by pressure. Consequently, it can only estimate hand pressure when all relevant visual features are visible in the camera image. In contrast, our approach doesn't require full hand visibility; as long as sufficient hand posture information can be inferred, pressure estimation is possible. Second, while we aim for comprehensive comparisons, PressureVision and PressureVision++ represent the only viable options available in current research for vision-based hand pressure estimation. While not a perfect comparison, we believe it offers valuable insights given the limited alternatives.
>
> **(W2-2) “The quantitative comparison between the two would be informative.”**
>
> Please refer (Joint Response 1).
>
> **(W3-1) Limitation on behavioral generalization to new objects and gestures**
>
> You correctly pointed out that our study does not demonstrate the performance of our framework on unseen gestures during the training phase. We acknowledge this limitation. While constructing our dataset, we strived to collect data encompassing a wide range of hand interactions, including plane interactions, pinch interactions, and grasp interactions, to facilitate comparisons with previous works [R1,R3]. However, we also had to consider the limited time and potential fatigue of our participants. Therefore, we made a selection of representative postures, as shown in Figure 8 and Figure 9. Collecting additional data for novel gestures with new participants is infeasible in this rebuttal phase, preventing us from conducting further experiments on generalization. However, as highlighted in Lines 908 and 152, our selection of postures covers a considerable portion of representative movements from the reference grasp taxonomy and those presented in [R3]. This suggests a degree of generalizability, but further research with new gestures and objects is certainly needed to solidify these points. Following this point, we will add this text to the Limitation section (Line 1107):
> - While our dataset covers a range of representative hand postures, it is limited in its exploration of novel gestures and object interactions unseen during training. Further research is necessary to investigate the generalizability of our framework to new objects and gestures. Expanding the dataset with a broader range of hand-object interaction scenarios and evaluating the model's performance on these unseen examples would provide valuable insights into its robustness and adaptability.
>
> **(W3-2) Clarification on matching the glove and vision-based hand pose estimation framework**
>
> You are correct, and we don't claim this to be a perfectly fair comparison. However, we'd like to provide some context on our rationale in two points. First, the authors of PressureVision++ specifically highlight its effectiveness in interacting with diverse surfaces (3rd, 4th, and 6th paragraphs of Introduction [R1]), even presenting training and data collection methodologies designed for such scenarios. This led us to believe that PressureVision++ would be a suitable candidate for comparison with our vision-based method. Second, when considering vision-based methods for comparison, we found PressureVision and PressureVision++ to be the only viable options available in existing research. While it may not meet your standards for a completely fair comparison, we believe we made the best possible choice given the available options.
>
> **(Q2) Why does utilizing hand angles not improve the model performance?**
> Based on the current results of our ablation study, we believe that hand angles alone lack sufficient inductive bias to effectively capture the spatial information inherent in hand postures.
>
> **(Q3) How was 3D hand pose acquired in cases without the glove?**
> We specify the utilization of an off-the-shelf hand pose detector in Section 5's introduction and Section 4.4, explicitly stating that we chose to employ ACR (Line 244). Additionally, as mentioned earlier, Section B.4.2, 3D Hand Pose, describes the canonicalization process for transforming the results of vision-based pose estimation into a standardized 3D hand pose representation (Lines 943-975).
>
> **(Q4) Can you show raw traces of the discretized pressure to give an intuition for what the real pressure data looks like?**
>
> To visually illustrate the performance of our model over time, we will include figures showing the temporal evolution of both ground truth and predicted pressure values. Figure R1 file provides an example of such visualization. The figure presents the GT pressure and the pressure predicted by our model for all 9 hand regions during TM-Press and Medium Wrap actions. The horizontal axis represents time, each row corresponds to a specific hand region, and the vertical axis in each plot shows the pressure values. In this example, the user performs the TM-Press for 10 seconds, followed by the Medium Wrap for another 10 seconds.  We believe that these temporal visualizations will enhance the readers' understanding of our framework by demonstrating how data is collected and predictions are made over time. We will include these figures and the corresponding discussion in a new Section D.4. Moreover, we’ll add one of those figures in the main body of the paper as in (Line 299) of Section 5.2.1.
>
> Due to limitations on the length of our rebuttal, we would like to address the minor points in separated comment.

---

> ### Author Response · Authors · 2024-08-07
> **Additional comment for minor points**
>
> **(M1) Datasets for Hand Pressure Estimation — do these datasets include pressure, many of these appear to be object interaction papers**
>
> You correctly pointed out that the line between object interaction papers and hand contact/pressure papers can be blurry, and Table 1 and Section 3.1, Vision-based Hand Pressure Estimation, reflect this. While some datasets focus solely on hand contact, others include pressure information, and some delve into object interaction to varying degrees. Rather than attempting a strict categorization, which can be challenging due to the inherent interconnectedness of these areas, we chose to provide a more comprehensive overview in this section. This approach allows for a broader discussion of related work and a better understanding of the research landscape.
>
> **(M2) Fig 4: can you clarify if this participant was in the training dataset?**
>
> Yes. Of course, this testing session on display does not overlap with the training session.
>
> **(M3) L302 ‘significant’ - were significance tests run here?**
>
> No. Thanks to Reviewer YYDP, we understand that the term ‘significant’ was understood differently for readers. We’ll remove that expression to avoid misunderstanding
>
> **(M3) Table 3 - what are the relevant error bars for this table?**
>
> Table R5 presents the same data as Table 3 in the main manuscript, but includes the standard deviation for each reported value. We will update Table 3 in the revised manuscript to include these standard deviations, providing a more complete picture of the variability in our results.
>
> **(M4) Table 2:unclear what errors are over**
>
> As guided in the Experiments section (Line 248), we present the evaluation metrics in Appendix D.2. To assess our model's performance, we utilize three metrics:  Coefficient of Determination (R²) which measures the proportion of variance in the actual pressure explained by our model, Normalized Root Mean Squared Error (NRMSE) which quantifies the precision of our model's pressure predictions, and Classification Accuracy, assessing the model's ability to correctly identify whether pressure is being applied to each region of the hand. We will add this summary to Line 248  of the main text.
>
> **(M5) Table 3: unclear what value is being plotted**
>
> The caption for Table 3 indicates that metric is NRMSE, but if there is still any ambiguity, please let us know again.
>
> **(M6) all exocentric?**
>
> Yes. In this research, we use an exocentric camera for camera modality. However, we believe it can be extended to usage of egocentric cameras if the hand pose estimation model is working well.
>
> **(M7) nit: Figure 3 pixelated**
>
> It’s a mistake. We’ll fix it.
>
> **(M8) Manuscript overstates at multiple points L316 "exceptional"**
>
> We’ll remove that expression to avoid overstatement. “Specific actions such as I-Press, M-Press, and R-Press exhibit high accuracy and low NRMSE,showing the model’s superior performance in simpler press interactions.”

---

> > ### Comment · Reviewer_YYDP · 2024-08-13
> > **Thank you for the detailed response**
> >
> > Thank you for the detailed point by point response, which addresses my original comments. I need more time to reconsider my original score but hopefully this will move it in a positive direction.
> >
> > One follow up on the Pressure Vision comparisons. These are great. I am wondering why the R2 is so bad. Is it predicting the pressure at the wrong time, is the pressure variable, or is it simply miscalibrated in intensity?

---

> > > ### Author Response · Authors · 2024-08-13
> > >
> > > Thank you for your thorough review and valuable contributions to our work. Your insights are greatly appreciated.
> > >
> > > Regarding the PressureVision++ quantitative comparisons, we found your comment about the low scores intriguing. To prevent miscalibration, we took steps to ensure that the units (psi, which is the unit outputted by the Sensel Morph pressure sensing array) and the range of values predicted by PressureVision++ were converted into Newton units, as is done in our paper, and calibrated for two different force ranges and units, enabling pressure estimation up to 20 N, as described in "B.4.1 sEMG Signal and Pressure Data" (Line 937 of our manuscript). This process helped mitigate potential miscalibration concerns, allowing for a fair comparison between the two systems. Moreover, we performed time synchronization between the camera and the Sensel Morph pressure sensing array. We verified the proper synchronization through data collection and preprocessing, ensuring no issues in this regard.
> > >
> > > However, it seems that PressureVision++ excels at detecting contact or the presence of force but faces challenges in accurately estimating force magnitude. This is consistent with the trends observed in PressureVision++ [R1]. For clarity, Table 2 (Performance compared to a PressureVision baseline) in [R1] primarily focuses on quantitative performance metrics of PressureVision++. Below is an excerpt from Table 2 in [R1]:
> > > | Method               | Contact Acc. | Contact IoU | Volumetric IoU |
> > > |----------------------|--------------|-------------|----------------|
> > > | PressureVision [R2]  | 72.7%        | 15.2%       | 11.3%          |
> > > | PressureVision++ [R1] | **89.3%**  | **41.9%**   | **27.5%**      |
> > >
> > > The primary performance metrics defined in [R1] are (1) Contact Accuracy, (2) Contact IoU, and (3) Volumetric IoU. Contact Accuracy and Contact IoU assess the correct identification of contact or the presence of force, corresponding to our "accuracy" metric. Conversely, Volumetric IoU reflects the prediction accuracy of pressure magnitude (section "5. Evaluation" of [R1]), corresponding to our R² or NRMSE, which represent regression task performance. While PressureVision++ demonstrates high performance in Contact Accuracy (measuring whether force presence is correctly identified regardless of location) and Contact IoU (IoU between the ground truth and predicted contact image), achieving 89.3% and 41.9%, respectively, Volumetric IoU only reaches 27.5%. This suggests that PressureVision++ excels in classification tasks, accurately identifying the presence or absence of force, but faces challenges in regression tasks, specifically estimating the precise force magnitude.
> > >
> > > This observation aligns with the trend we observed in our Table R2. In Table R2, PressureVision++ shows comparable accuracy (measuring whether force presence is correctly identified for all five fingers) to the Force-aware Interface (67.90% versus 66.00%). However, PressureVision++ exhibits relatively low values in R-square (40.30%) and NRMSE (32.95%), indicating less accurate performance in estimating force regression. Based on these results, PressureVision++ demonstrates strength in detecting contact or the presence of force but struggles to accurately estimate force magnitude. In comparison, the sEMG only model, while exhibiting similar performance in accuracy to PressureVision++, seems to have a comparative advantage in accurately estimating the precise force magnitude.
> > >
> > > Furthermore, to ensure transparency and reproducibility of our evaluation of PressureVision++, we will publicly release all data and code used for quantitative assessment and performance calculation of comparative methodologies. This information will be included in a new section, "Cross-User Evaluation," in Section D (Line 1046-1047) of the revised manuscript.

---

### Official Review · Reviewer_nF88 · 2024-07-12

**Soundness:** 3
**Presentation:** 3
**Contribution:** 3
**Rating:** 5
**Confidence:** 4

**Summary:**

This paper proposes a hand pressure estimation method based on 3D hand poses and t forearm surface electromyography (sEMG) signals. Accordingly, the paper constructs a multimodal dataset containing pressure, 3D hand poses, and sEMG signals. The paper experimentally verifies that combining 3D hand poses and sEMG signals achieves better results compared to using these two modalities independently. Furthermore, compared to vision-based pressure estimation methods, the method achieves better results in complex hand-object interaction scenarios.

**Strengths:**

1) The paper is well written and easy to follow.
2) The idea of integrating 3D hand poses and electromyography signals for hand pressure estimation in this paper is intuitive, interesting, and practical. The experiments in this paper further demonstrate that combining hand poses and electromyography signals is beneficial.
3) The data collection system constructed in this paper is impressive, as collecting high-quality synchronized multimodal hand data is very challenging. This dataset could potentially advance the field of hand pressure perception.

**Weaknesses:**

1) From the perspective of multimodal fusion methods, the framework proposed in this paper lacks novelty. This framework directly combines the global features of the electromyography modality and the global features of hand poses without fully utilizing the fine-grained spatial information and prior structural information of the hand poses.

2) The qualitative comparison with PressureVision++ is unfair. Due to biases in the training dataset and the design preferences of the method, PressureVision++ is more suitable for pressure estimation in planar contact scenarios. However, this paper does not compare with PressureVision++ in such scenarios.

3) The training and testing setups are not sufficiently reasonable. During training, hand poses captured by a data glove are used as input, while during testing, the estimated results from a vision-based hand pose estimation model are used as input. There is a significant gap in the distribution of hand poses, which may potentially impair the performance of the hand pose branch.

**Questions:**

1. Would it be beneficial for the overall method's performance to use predicted hand poses as input during training?
2. Have you tried other multimodal information fusion methods?

**Limitations:**

Yes

---

> ### Author Rebuttal · Authors · 2024-08-07
>
> **(W1) Clarifications on the novelty of the proposed framework**
>  - We acknowledge that there might be room for improvement in terms of methodological novelty. In this paper, however, our primary objective of this paper was to pioneer the use of electromyography and 3D hand posture data simultaneously and demonstrate its efficacy. To achieve this, we presented a data collection platform and a pipeline for using our model in inference without a data glove. Subsequently, to validate the approach experimentally, we deliberately opted for the simplest form of machine learning modeling approach instead of focusing on introducing a new model architecture by incorporating geometric structures of the model and data, our work lays a strong foundation for future research in this direction. We will add this point in the Limitations and Future Work section (Line 1107) as follows:
> - “While our current framework demonstrates the value of combining sEMG and 3D hand pose data, it primarily focuses on leveraging global features from both modalities. Future work could explore incorporating fine-grained spatial information and prior structural knowledge of hand poses into the model architecture to further enhance pressure estimation accuracy and robustness.”
>
> **(W2) Clarification on the comparison with PressureVision++**
> - We agree that this is not a perfectly fair comparison. Regarding this, we would like to clarify the rationale for our comparison and explain the additional quantitative comparison with PressureVision++ we carried out during our rebuttal.
> Unlike PressureVision, the authors of PressureVision++ specifically highlighted its effectiveness in interacting with diverse surfaces (3rd, 4th, and 6th paragraphs of Introduction in [R1]), even presenting training and data collection methodologies designed for such scenarios. This led us to believe that PressureVision++ would be a suitable candidate for comparison with our vision-based method, which is why the primary reason we chose PressureVision++[R1] over PressureVision[R2].
> When considering vision-based methods for comparison, we found PressureVision and PressureVision++ to be the only viable options available in existing research. While it may not meet your standards for a completely fair comparison, we believe we made the best possible choice, given the available options.
> We carried out a quantitative comparison with PressureVision++ to support our claim with a fairer condition.
>
> **(W3) Clarification on utilizing vision-based hand pose estimation model for testing setup**
> - As outlined in Section 4.4 (Line 228), we canonicalize 3D hand pose data derived from a vision-based hand pose estimator (More details shown in Section B.4.2, Line 942). During inference, we align the 3D hand pose estimated from visual data with the training data representation. This process involves rescaling the hand pose to match the bone lengths used during training, rotating it to align with the kinematic tree of the MANO hand model, and translating it to ensure consistent root joint positioning. This canonicalization minimizes the discrepancy between hand pose data from the glove and the vision-based estimator, improving the generalizability of our model during inference. Also our framework inherently relies on the accuracy of off-the-shelf hand pose detectors during inference (Line 1097). We acknowledge that this dependency is a limitation of the proposed modality and inference pipeline and discuss this limitation in Section E. Limitations and Future Works (Line 1077).
>
> **(Q1) Benefits of utilizing predicted hand poses as input for training**
>  - While using predicted hand poses during training might seem beneficial, our training dataset requires participants to wear a glove to collect the ground truth hand pressure data. We utilize a pressure-sensing glove (TactileGlove, Pressure Profile Systems) for this purpose. Unfortunately, most current hand pose detectors are not trained to work on hands wearing gloves, or their performance significantly degrades in such situations. Therefore, it is hard to incorporate predicted hand poses during the training phase due to the need for more reliable hand pose estimation for gloved hands.
>
>
> **(Q2) Have you tried other multimodal information fusion methods?**
>  - We haven't explored alternative multimodal information fusion methods in this work. However, we acknowledge that there are numerous potential improvements to be made in both the model design and implementation.  Investigating different fusion techniques is a promising direction for future work, and we believe it holds the potential to enhance the accuracy and robustness of our hand pressure estimation framework. We will add this point into limitation and future works section as follows:
> “While our current framework demonstrates the value of combining sEMG and 3D hand pose data, it primarily focuses on leveraging global features from both modalities. Future work could explore incorporating fine-grained spatial information and prior structural knowledge of hand poses into the model architecture to further enhance pressure estimation accuracy and robustness.”

---

> > ### Comment · Reviewer_nF88 · 2024-08-12
> >
> > I have carefully read the author's meticulous rebuttal and the reviews of other reviewers. I thank the authors for providing a detailed rebuttal. First, my concerns about comparison with PressureVision++  have been addressed. Although the authors state that their main contribution is not the multimodal fusion method, I still think it is important. After all, it is “foreseeable ” that fusing more modes can improve the accuracy of estimation. Overall, considering the pioneering nature and adequacy of the experiments, I maintain my initial score, which is borderline accept, but there are still some weaknesses in the depth and innovation of the fusion framework.

---

> > > ### Author Response · Authors · 2024-08-12
> > >
> > > We sincerely appreciate you taking the time to thoroughly review our rebuttal and acknowledge the clarification regarding our comparison with PressureVision++. We understand your perspective on the importance of multimodal fusion methods and agree that there is a clear potential for improving our approach in that aspect. We are grateful for your constructive feedback and insightful suggestions, particularly regarding the exploration of fine-grained spatial information and prior structural knowledge of hand poses.
> > >
> > > Thank you again for your valuable contribution to our work.

---

### Official Review · Reviewer_3PTe · 2024-07-14

**Soundness:** 2
**Presentation:** 2
**Contribution:** 3
**Rating:** 4
**Confidence:** 4

**Summary:**

This paper presents a novel framework for estimating hand pressure during various hand-object interactions using multimodal data. The framework integrates sEMG and 3D hand posture information to enhance the accuracy of pressure estimation. They introduce a dataset to validate their approach. The primary contribution of the study is the exploration of combining vision-driven 3D hand posture data with sEMG to improve the robustness of hand pressure estimation.

**Strengths:**

1. The integration of sEMG signals with 3D hand posture data sound good.

2. The dataset is extensive, including 83.2 million frames from various hand-object interactions.

3. The proposed model shows high accuracy in estimating hand pressure.

**Weaknesses:**

1. Although integrating sEMG signals with 3D hand posture data is a promising approach, there is insufficient experimental evidence to support the motivation behind this combination. The paper lacks clear proof of the benefits described, beside the sentence.

2. The paper does not adequately compare their approach with other vision-based datasets or techniques that also extract 3D hand posture information.

3. While combining sEMG and vision-based methods theoretically addresses individual robustness issues, the claim that this combination improves overall robustness is not convincingly supported by the data. The dataset should demonstrate the robustness of the proposed method, not just improvements in accuracy.

**Questions:**

Please list up and carefully describe any questions and suggestions for the authors. Think of the things where a response from the author can change your opinion, clarify a confusion, or address a limitation. This is important for a productive rebuttal and discussion phase with the authors.

**Limitations:**

see weaknesses

---

> ### Author Rebuttal · Authors · 2024-08-07
>
> **(W1) Support for advantages of integrating sEMG signals with 3D Hand Posture data**
>    - We would like to clarify that we performed an ablation study (Table 2 and 3) and discussed study results in the corresponding sections. These tables compare the performance of our model using: (1) sEMG data only[R3], (2) sEMG data with hand posture data using angle representation, and (3) sEMG data with 3D hand pose data(Ours). We primarily discussed this analysis in Section 5.2.1 highlighting the benefits of combining sEMG and 3D hand pose information.
> We acknowledge that our initial explanation of the motivation might not be sufficiently convincing for readers. To provide stronger empirical motivation for our approach, we will add a new figure illustrating cases where similar sEMG patterns are observed despite pressure being applied to different parts of the hand. Figure R4 demonstrates this phenomenon. The figure displays the time-series representation of 8-channel sEMG signals over 5 seconds for two pairs of hand postures: (1) I-Press and M-Press, (2) TI-Pinch and TM-Pinch. Each row represents a different sEMG channel. In both pairs, the pressure is applied differently, targeting either the index fingertip or the middle fingertip. However, we observe very similar sEMG patterns between I-Press and M-Press, and between TI-Pinch and TM-Pinch.
> This observation demonstrates that while sEMG signals provide valuable information about force, they may not be sufficient for fine-grained pressure localization on their own. This highlights the potential of incorporating hand posture information to guide pressure estimation. We will incorporate this discussion and Figure R4 into the revised manuscript, adding a new second paragraph in Section 4.1 (Llines 172-173) and a new Appendix section titled "Empirical motivation of our framework" to further clarify the rationale behind our approach.
>
> **(W2) Comparison with vision-based datasets/techniques extracting 3D hand posture info**
>    - We agree that we did not comprehensively cover various vision-based datasets or methodologies for 3D hand estimation as related works or comparative methodologies. However, please take it into consideration that the ultimate goal of our research is to estimate hand pressure, not to derive 3D hand posture. For this reason, we selected comparative methodologies and datasets aimed at estimating hand contact or hand pressure rather than 3D hand posture estimation. Since the ultimate goal of our research is hand pressure estimation, we believe that comparisons with 3D hand pose estimation methodologies and datasets may not be contextually consistent. However, we also found out that we missed highly related dataset paper such as ActionSense from NeurIPS 2022 which will be added as part of our related works.
>
> **(W3) Clarification on the meaning of the robustness in our paper**
> - In our manuscript, the 'robustness' of the model refers to the model’s capability to ensure high accuracy across various postures over different hand regions. Here, we have detailed the performance across different hand regions (Table 3) for various postures (Figure 3) to show our model’s robustness. Thanks to Reviewer 3PTe, we understand that the term 'robustness' was understood differently by readers, we will revise it to clearly convey our intended meaning. Specifically, we will revise the following lines as follows:
>
> - (Line 313) “This consistent correlation between predicted values and actual pressure measurements highlights the model’s robustness and adaptability in handling a wide range of hand movements.”
> → “This consistent correlation between predicted values and actual pressure measurements highlights the model’s ability to maintain high accuracy and reliability across a diverse range of hand parts and postures, thereby demonstrating its robustness.”
> - (Line 331) “Figure 4b shows demo video footage illustrating our framework’s robustness in estimating hand pressure while continuously changing hand posture, pressure levels, and the objects being grasped.”
> → “"Figure 4b shows demo video footage illustrating our framework’s capability in accurately estimating hand pressure while continuously changing hand posture, pressure levels, and the objects being grasped."

---

> > ### Author Response · Authors · 2024-08-14
> >
> > We deeply appreciate your meticulous review of our manuscript and the insightful suggestions provided. Your feedback has been instrumental in guiding our revisions, and we have diligently addressed the remaining points raised in your comments. As the discussion phase concludes, we are confident that the detailed clarifications and newly suggested empirical motiation, as presented in our rebuttal, provide a compelling testament to the effectiveness of our proposed method. We sincerely hope that these revisions have adequately resolved your concerns. We remain available to address any further questions you may have and are grateful for your valuable contribution to improving the quality of our work.

---

### Author Rebuttal · Authors · 2024-08-07

We would like to thank reviewers for the constructive feedback.

We have been putting our best effort to address the weaknesses and questions pointed by reviewers to strengthen the paper. Our rebuttal PDF includes new experimental results, diagrams, and tables to support our work. In our rebuttal comments, tables and figure  in the rebuttal PDF are referenced as Figure Rx & Table Rx while figures and tables in the main paper are denoted as Figure x and Table x. Below, we have listed the references used in our rebuttal.

We summarize the two major changes and updates we have made during the rebuttal period:
### **(Joint Response 1) Quantitative comparison with vision-based method (PressureVision++[R1])**
We appreciate the reviewer's suggestion for a quantitative comparison with vision-based methods. Yes, while PressureVision methods are inherently limited in their application due to the necessity of a glove for whole-hand pressure measurement, a quantitative comparison would provide valuable insights.

To address this, we conducted a quantitative evaluation of PressureVision++ using the same equipment as the original PressureVision++ study: a Logitech Brio 4k webcam and a Sensel Morph pressure sensing array. As PressureVision++ estimates pressure only on fingertips and requires full visibility of the hand within the camera view, we focused our evaluation on plane and pinch interaction sets, specifically: I-Press, M-Press, R-Press, P-Press, IM-Press, MR-Press, TI-Pinch, TM-Pinch, TIM-Pinch, TIMR-Pinch, and TIMRP-Pinch. Following our data collection protocol, each participant was instructed to repeat each action for 30 seconds. To ensure optimal visibility for PressureVision++, we carefully adjusted the camera angle to capture both the fingers and palm. Figure R2 and Figure R3 illustrate the data collection setup for PressureVision++. We collected data from 5 participants using this setup, mirroring the methodology of PressureVision++. Please note that the Sensel Morph could not measure thumb force during pinch actions, so this data was excluded from the analysis. Therefore, Table R2 and Table R4 present the performance comparison between PressureVision++ and our model, focusing on the pressure exerted by the tips of the index, middle, ring, and pinky fingers for the specified action set. It is important to note that our quantitative evaluation of PressureVision++ is inherently a cross-user performance report, as the model was trained on a separate dataset. For fairness and consistency, we report the cross-user performance for "sEMG Only[R3]" and "sEMG + 3D Hand Posture(Ours)" in Table R2 and Table R4 as well. Table R2 provides a comparison across all action sets for the three main metrics, while Table R4 offers a detailed breakdown of fingertip performance for plane interactions and pinch actions. As expected, there is a general decrease in performance when moving from within-user to cross-user evaluation. For instance, in our method, we observe a decrease in R² from 88.86% to 66.71%, in NRMSE from 6.65% to 9.27%, and in accuracy from 83.17% to 82.20%. At the same time, as shown in Tables R2 and R4, our method still significantly outperforms both PressureVision++ (66.60% in accuracy) and the sEMG only model (67.90% in accuracy) in the cross-user evaluation. This indicates the superiority of our method in both within-user and cross-user scenarios.

### **(Joint Response 2) Cross-user Evaluation**
We acknowledge the reviewer's concern regarding the limited within-participant evaluation and the potential variability of sEMG signals across individuals. To address this, we conducted additional experiments in a cross-user setting to measure the generalizability of our model. Specifically, we used all session data acquired from 16 participants as our training set, while data from the remaining 4 participants was allocated to the test set. This ensures that the model is completely blind to the test participants during both training and evaluation. Table R1 and R3 present the results of this cross-user evaluation, comparing our model's performance against the primary baseline method Force-aware interface[R3]. Furthermore, in the rebuttal PDF file, Table R2 and Table R4 provide a breakdown of our performance for index, middle, ring, and pinky finger pressures during plane and pinch interactions, specifically for comparison against PressureVision++. The results from these cross-user evaluations demonstrate that our approach consistently outperforms existing methods[R1,R3] even when encountering unseen participants. For a more detailed analysis of our performance compared to PressureVision++, please refer to our response regarding the quantitative evaluation of PressureVision++ below. We will add new results and discussion as a new subsection titled “Cross-User Evaluation” in Section D (Line 1046-1047) of the revised manuscript.

Please refer to reviewer-specific feedback and a one-page PDF with a summary of added experimental results.

### References

- [R1] Grady, Patrick, et al. "PressureVision++: Estimating Fingertip Pressure from Diverse RGB Images.", CVPR 2024.
- [R2] Grady, Patrick, et al. "PressureVision: estimating hand pressure from a single RGB image." ECCV 2022.
- [R3] Zhang, Yunxiang, et al. "Force-aware interface via electromyography for natural VR/AR interaction." ACM Transactions on Graphics (TOG) 2022.
- [R4] Lalitharatne, Thilina Dulantha, et al. "A study on effects of muscle fatigue on EMG-based control for human upper-limb power-assist." ICIAfS 2012.
- [R5] Eddy, Ethan, Erik J. Scheme, and Scott Bateman. "A framework and call to action for the future development of EMG-based input in HCI." ACM CHI 2023.

---

### Comment · Area_Chair_vRBg · 2024-08-12
**Please check the author rebuttal**

Dear Reviewers of submission 7975,

Thank you for your time and effort so far in reviewing this paper. The author kindly provided the rebuttal to your comments, and the deadline for the author-reviewer discussion is tomorrow (13th August). It becomes more critical for this paper since reviews of this paper is mixed. It would be much appreciated if you could participant in the discussion with author to check if your concerns are addressed by the author or not. Thank you!

Best regards,

Your AC of submission 7975.

---

### Decision · Program_Chairs · 2024-09-25

**Decision:**

Accept (poster)

**Comment:**

The reviewers appreciate the collected dataset and the idea of combining 3D hand poses and electromyography signals for hand pressure estimation. The reviewer raised concerns about the lack of comparison with other vision-based baselines, and novelty in the data fusion framework, etc. The author provided detailed and professional rebuttals to address most of the concerns. After reading these comments, I give my suggestion as acceptance of this paper.